



# Evaluating seasonal and regional distribution of snowfall in regional climate model simulations in the Arctic

Annakaisa von Lerber[1,2], Mario Mech[1], Annette Rinke[3], Damao Zhang[4], Melanie Lauer[1],
Ana Radovan[5], Irina Gorodetskaya[6], and Susanne Crewell[1]

[1]University of Cologne, Cologne, Germany
[2]Finnish Meteorological Institute, Helsinki, Finland
[3]Alfred Wegener Institute, Helmholtz Centre for Polar and Marine Research, Potsdam, Germany
[4]Pacific Northwest National Laboratory, Washington, USA
[5]Deutscher Wetterdienst, Germany
[6]University of Aveiro, Aveiro, Portugal

**Correspondence:** Annakaisa von Lerber (Annakaisa.von.Lerber@fmi.fi), Susanne Crewell, susanne.crewell@uni-koeln.de

**Abstract.** In this study, we investigate how the regional climate model HIRHAM5 reproduces the spatial and temporal distribution of Arctic snowfall when compared to CloudSat satellite observations during the examined period of 2007 - 2010. For this purpose, both approaches, i.e. the assessment of surface snowfall rate (observation-to-model) and the radar reflectivity factor profiles (model-to-observation), are carried out considering spatial and temporal sampling differences. The HIRHAM5

model, which is constrained in its synoptic representation by nudging to ERA-Interim, represents the snowfall in the Arctic region well in comparison to CloudSat products. The spatial distribution of the snowfall patterns is similar in both identifying the southeastern coast of Greenland and the North Atlantic corridor as regions gaining more than twice as much snowfall as the Arctic average, defined here for latitudes between 66°N and 81°N. An excellent agreement (difference less than 1%) in Arctic averaged annual snowfall rate between HIRHAM5 and CloudSat is found whereas ERA-Interim reanalysis shows an

underestimation of 45% and significant deficits in the representation of the snowfall frequency distribution. From the spatial analysis it can be seen that the largest differences in the mean annual snowfall rates are an overestimation near the coastlines of Greenland and other regions with large orographical variations, as well as an underestimation in the northern North Atlantic ocean. To a large extent, the differences can be explained by clutter contamination, blind zone or higher resolution of CloudSat measurements, but clearly HIRHAM5 overestimates the orographic-driven precipitation. The underestimation of HIRHAM5

within the North Atlantic corridor south of Svalbard is likely connected to a poor description of the marine cold air outbreaks which could be identified by separating snowfall into different circulation weather type regimes. By simulating the radar reflectivity factor profiles from HIRHAM5 utilizing the PAMTRA forward-modeling operator, the contribution of individual hydrometeor types can be assessed. Looking at a latitude band at 72 - 73° N, snow can be identified as the hydrometeor type dominating radar reflectivity factor values across all seasons. The largest differences between the observed and simulated re-

flectivity factor values are related to the contribution of cloud ice particles, which is underestimated in the model most likely due to the small size of the particles. The model-to-observation approach offers a promising diagnostic when improving cloud schemes as illustrated by comparison of different schemes available for HIRHAM5.



# 1 Introduction

Globally, precipitation acts as a significant coupling between Earth's hydrological, energy and bio-geochemical cycles (Hou
et al., 2014) and therefore, snowfall is an important climate indicator. For example, snowfall affects seasonal growth and
decay of sea ice in the Arctic by accumulating on ice (Screen and Simmonds, 2012; Merkouriadi et al., 2017; Sato and Inoue,
2018; Webster et al., 2018). Further, it contributes to the freshwater input into the ocean (Prowse et al., 2015; Vihma et al.,
2016), modulates the surface albedo (Box et al., 2012; Riihelä et al., 2019), and is the primary source of mass for the ice
sheets e.g. on Greenland  (van den Broeke et al., 2009) or East Antarctic Ice Sheet (Boening et al., 2012). However, it is
still one of the most uncertain variables in numerical weather prediction (NWP) models as well as in climate simulations and
reanalyses (Boisvert et al., 2018; Behrangi et al., 2016). As shown by Boisvert et al. (2018) when comparing mean cumulative
annual snowfall among eight different reanalyses over the Arctic Ocean during the years of 2000-2016, the standard deviation
between the products can be 60-70 mm in yearly mean rate, which is about half of the total snowfall rates estimated by some
reanalyses products. Thus, comprehensive observations and modeling simulations are required to increase our understanding of
the seasonal and regional snowfall patterns and how these are dependent on the large-scale atmospheric circulation. However,
it is challenging to capture snowfall at the relevant scales both in observations and in models (Tapiador et al., 2017).

Because cloud microphysical processes act on rather small scales they need to be parameterized in atmospheric models.
Modelling of cloud microphysics has improved during recent years with more complex approaches and increased higher
resolution (Grabowski et al., 2019). Still in most climate models, precipitation is a diagnostic variable and reanalyses do
not assimilate observations of precipitation (Boisvert et al., 2018; Knudsen et al., 2018). Thus, solid precipitation is solely
determined by the model and subject to large uncertainties (Kalnay et al., 1996). For example, the representation of Arctic
mixed-phase clouds for snowfall is important, but still a challenge for models (Morrison et al., 2012; McIlhattan et al., 2017;
Sedlar et al., 2020). Regional Climate Models (RCMs) can provide both high spatial and temporal resolution (few kilometers
and hourly, respectively) in areas with little or no observational data. This makes them useful for evaluating climate at the
local scale and in areas with sparse ground observations (Silverman et al., 2013). In particular, we need to assess their skills in
precipitation simulation in order to be able to investigate changes of snowfall in future climate.

Model performance has been assessed not only between different models and reanalyses, but also to observations, either
ground-based or space-borne as, e.g. in Lindsay et al. (2014) to monthly mean values of automatic and manual gauges from the
limited land stations, or in  Boisvert et al. (2018) to drifting ice mass balance buoys. Palerme et al. (2017) and Edel et al. (2020)
compared snowfall climatologies from CloudSat radar observations (Stephens et al., 2008) to reanalyses for Antarctica and
the Arctic. In situ instruments such as gauges and disdrometers are sparse in the Arctic area, suffer from the biases introduced
by blowing and drifting snow and show generally an underestimation of snowfall under windy conditions (Goodison et al.,
1998; Wolff et al., 2012; Rasmussen et al., 2012). Even more sparse are sites with extensive ground-based remote sensing
instrumentation such as cloud radars and radiometers, which provide anchor points for process understanding and validation
(e.g., Castellani et al., 2015; Verlinde et al., 2016; Maturilli et al., 2013; Pettersen et al., 2018; Nomokonova et al., 2019;
Gierens et al., 2020; Schoger et al., 2021).





CloudSat has been the only satellite, which has onboard a microwave radar with sufficient sensitivity that reaches higher latitudes to give accurate snowfall estimates for the Arctic region (Stephens et al., 2008; Kidd and Huffman, 2011). CloudSat data have been widely used in model comparisons, (e.g. Hiley et al., 2011; Palerme et al., 2014; Kulie et al., 2016; Palerme et al.,

2017; Souverijns et al., 2018; Milani et al., 2018; Adhikari et al., 2018; Edel et al., 2020), and the derived snowfall climatology has shown good agreement with ground-based radar or in situ observations both in Arctic and Antarctic (Souverijns et al., 2018; Bennartz et al., 2019; Kodamana and Fletcher, 2021; Duffy et al., 2021).

CloudSat has its limitations with a narrow swath and a long revisiting time of 16 days. Thus, the defined snowfall climatology is dependent on the used sampling grid. Choosing a coarse spatial resolution, which will increase the number of samples per grid

point, and smoothing the strong peak values while averaging, will basically lead to an underestimation of the total snowfall rate. Therefore, the uncertainties in snowfall rates induced from the low temporal resolution of CloudSat should be compensated with the high spatial resolution of the observations in comparison to spatially coarse-resolution model values. The poor fractional coverage might be less of an issue at the high latitudes where convective precipitation is seldom and precipitating systems mostly occur at large scale, except for some small scale orographic precipitation (Palerme et al., 2014). Souverijns et al. (2018)

showed that despite the long revisiting time, in re-sampling the surface snowfall data to a 1° latitude by 2° longitude grid, the snowfall climatology is represented with reasonable accuracy of 15 % in the Antarctic region when compared to three ground station observations. Edel et al. (2020) composed the snowfall climatology based on CloudSat observations with a similar sampling grid over the years 2007 - 2010 and compared the frequency and phase of precipitation to modeled values of ERA-Interim and two versions of the Arctic System Reanalysis (ASR), finding similar geographical patterns but also significant

mean snowfall rate differences, especially over Greenland. Thomas et al. (2019) found considerable differences in the statistical distributions of different climate models judged against Cloudsat illustrating the need for further model improvement.

Typically, space-borne active measurements suffer from ground clutter. With CloudSat, it is assumed that observations over the land areas below 1000 m and over the sea below 500 m may suffer from ground clutter contamination and are typically discarded from the analysis (Palerme et al., 2019). Therefore, the discarded so-called blind zone may cause an underestimation

of the surface snowfall rate (about 10%) as the microphysical growth processes in snow can significantly enhance the snowfall intensity near the surface (Maahn et al., 2014). Another limitation of utilizing remote sensing observations to evaluate snowfall rate is the uncertainty of the used retrieval that derives the rate from the measured radar reflectivity factor (e.g., Kulie and Bennartz, 2009; Milani et al., 2018). However, while uncertainties on individual precipitation retrievals from CloudSat data may potentially be large, the mean uncertainty should be much smaller (Palerme et al., 2014).

In this study, we evaluate the performance of the HIRHAM5 RCM  (Christensen et al., 2007) to reproduce the seasonal and regional distribution of Arctic snowfall by comparison to CloudSat observations. To consider the above-mentioned pitfalls when comparing the model estimates with space-borne radar observations, we adopted two approaches: (i) observation-to-model and (ii) model-to-observation. In (i), the surface snowfall rate modeled by the RCM is compared to the retrieved surface snowfall rate from the CloudSat measurements similarly as in the studies mentioned above. In (ii), the RCM output is fed

into a forward simulator and the assessment is performed by comparing the simulated profiles of radar reflectivity factor with the observed ones. HIRHAM5 output including thermodynamic state and mixing ratios of different hydrometeors is





inserted into the Passive and Active Microwave TRAnsfer tool (PAMTRA; Mech et al., 2020) to compute attenuated and unattenuated reflectivity factor profiles. The similarities and differences are investigated for the different Arctic regions and seasons separately to clarify how well the HIRHAM5 models the processes related to snowfall.

The paper is structured as follows. The next section briefly introduces the data sources: the RCM HIRHAM5, forward simulator PAMTRA, and CloudSat observations. The Sect. 3 illustrates the methodology how to sample the data sets for a fair comparison and introduces the circulation weather type (CWT) diagnostic, whereas the detailed description how the HIRHAM5 output is converted for PAMTRA calculations is outlined in Appendix A. Sects. 4 and 5 present the results, firstly with the approach to assess the surface snowfall rate and secondly the comparison in the modeled and measured reflectivity

factor regime. The seasonal and spatial differences are discussed and the conclusions and future aspects are summarized in Sect. 6. To simplify the text, from now on in this study, the reflectivity factor is described simply as reflectivity.

## 2   Data

To study the regional differences, the Arctic region is divided into twelve different areas (Fig. 1 and Table 1), covering the latitudes between 66°N and 81°N. The studied region is restricted on one hand by the size of the HIRHAM5 domain and on

the other hand by the CloudSat coverage. For the twelve areas, the region is distributed to 60°sectors in longitude, and in latitude to two rings, covering the 66°N to 70°N and 70°N to 81°N. The two rings are separated to clarify the different characteristics of the southern and northern regions. The studied period is between years 2007-2010, defined from the availability of an all-day period of CloudSat data.

### 2.1   HIRHAM5

HIRHAM RCM is based on the dynamics of the High Resolution Limited Area Model (*HIR*LAM; Undén et al., 2002) and the physical parametrizations of the atmospheric general circulation model EC*HAM* (Roeckner et al., 2003). We utilize here the version 5, which combines HIRLAM release 7.0 with ECHAM model release 5.2.02 (Christensen et al., 2007). HIRHAM5 has been applied to various Arctic studies (recently, e.g., Akperov et al., 2019; Sedlar et al., 2020; Inoue et al., 2021).

    HIRHAM5 is run at a horizontal resolution of 0.25° (about 27 km) on a rotated latitude-longitude grid with the North Pole

on the geographical equator at 0° E, and with 40 vertical levels. The altitude ranges from about 10 m above the surface up to 10 hPa and the lowermost 1 km is represented by 10 levels. In this study, the daily snowfall rate constructed from the 3-hourly output is used for the observation-to-model approach. On the other hand, the 3-hourly output of the thermodynamic state and the mass mixing ratios of cloud ice, cloud liquid, snow, and rain is used for calculating synthetic reflectivities in the model-to-observation approach. Whereas cloud ice and liquid mixing ratios are prognostic variables in the model, snow and rain are

diagnostic variables.

    We apply at every time step a grid point nudging, i.e. dynamical relaxation (sometimes also called indiscriminate nudging or grid relaxation), which was originally developed for data assimilation (Omrani et al., 2012). We have chosen a parameter value corresponding to 1% nudging. This ensures to constrain the simulated large-scale flow to the driving ERA-Interim



**Table 1.** The studied twelve regions divided equally in the longitude with 60° to have same sampling size between the different regions and in latitude divided into two latitude rings to examine the southern and northern parts of the Arctic separately. The last column states statistically coinciding HIRHAM5 model run times with the CloudSat overpasses.

| Nr | Lon | Lat | Region | HIRHAM5 runs [UTC] |
|---|---|---|---|---|
| 1 | 20°W - 40°E | 66°N - 70°N | Arctic North Atlantic | 03, 09, 12 |
| 2 | | 70°N - 81°N | | 03, 09,12 |
| 3 | 40°E - 100°E | 66°N - 70°N | Kara Sea | 00, 06, 09, 21 |
| 4 | | 70°N - 81°N | | 00, 06, 09, 21 |
| 5 | 100°E - 160°E | 66°N - 70°N | Laptev Sea | 03, 06, 18 |
| 6 | | 70°N - 81°N | | 03, 18, 21 |
| 7 | 160°E - 140°W | 66°N - 70°N | Chukchi Sea and Beaufort Sea | 00, 12, 15 |
| 8 | | 70°N -81°N | | 00, 15, 21 |
| 9 | 140°W - 80°W | 66°N - 70°N | Canadian archipelago | 09, 12, 15 |
| 10 | | 70°N - 81°N | | 09, 12, 21 |
| 11 | 80°W - 20°W | 66°N - 70°N | Greenland and Baffin Bay | 06, 09, 15, 18 |
| 12 | | 70°N - 81°N | | 06, 09, 15 |

reanalysis. For an evaluation of HIRHAM5 snowfall using observations, the simulations must stay reasonably close to the real

development of the synoptic weather situation allowing to separate dynamical from microphysical effects. Due to the nudging, the large-scale snowfall patterns are expected to correspond with reasonable accuracy to the observations, as is demonstrated in a snowfall case study for March 7, 2010 in Figs. 2 and A1. The location of the precipitation system and its vertical extent is rather similar in HIRHAM5 and CloudSat observations. Therefore, it is assumed that the differences between the modeled snowfall and observations are mostly caused by the ECHAM5 microphysical parameterization employed in HIRHAM5 and

observational uncertainties.

Unless specified, HIRHAM5 uses the modified Tompkins cloud scheme (Klaus et al., 2016) throughout the whole paper. However, to study the effect of different cloud cover schemes, we have run the model for the studied period also with two other schemes: the original Tompkins (Tompkins, 2002) and the Sundqvist (Sundqvist et al., 1989) cloud schemes. The Sundqvist scheme is based on relative humidity, i.e. a critical threshold of relative humidity controls the cloud cover formation. The

threshold decreases exponentially from 90% near the surface to 70% at higher altitudes. The Tompkins scheme is a prognostic statistical cloud scheme. The subgrid-scale variability of total atmospheric water content is specified by a probability density function in terms of the beta distribution. The higher-order moments of the beta distribution, namely variance and skewness, are included and linked to subgrid-scale processes like turbulence, convection, and microphysics. Fractional cloud cover is





computed as an integral over the supersaturation part of the actual beta distribution. The Tompkins scheme includes adjustable

parameters, which determine the shape of the beta distribution and microphysical processes (e.g., the aggregation rate - the efficiency of snow formation by aggregation of cloud ice particles, etc.). Klaus et al. (2016) found that a parameter tuning of the cloud ice threshold controlling the efficiency of the Bergeron-Findeisen process, combined with a scheme extension which allows negatively skewed beta distributions is most suitable for Arctic cloud simulations. They showed that this modified Tompkins scheme significantly reduces Arctic cloud cover being in better agreement with CloudSat/CALIPSO observations.

The more efficient Bergeron-Findeisen process decreases (increases) the cloud water (ice) content.

## 2.2 ERA-Interim

ERA-Interim is a global atmospheric reanalysis produced by the ECMWF (Dee et al., 2011). It was widely used as a reference, covering the period from 1 January 1979 onward until to 31 August 2019 and was replaced by the ERA5 reanalysis also available from 1 January 1979 onward. The system includes a 4-dimensional variational analysis (4D-Var) with a 12-hour

analysis window. The spatial resolution of the data set is approximately 80 km (T255 spectral) on 60 levels in the vertical from the surface up to 0.1 hPa (Dee et al., 2011). To improve and constrain the forecast, surface, radio-sounding, and airborne observations, as well as satellite measurements, are assimilated into the ERA-Interim (Dee et al., 2011). However, CloudSat observations are not applied, nor are the direct precipitation observations from any source. The cloud microphysics scheme utilized in ERA-Interim is based on Tiedtke (1993) representing clouds in terms of two prognostics variables. One variable is

for cloud fraction and the other one for total cloud condensate, which in turn is divided into separate liquid and ice categories diagnostically according to temperature (Forbes et al., 2011). ERA-Interim is used here as lateral forcing as well as for the nudging for the HIRHAM5 simulations. Comparing both HIRHAM5 and ERA-Interim surface snowfall rates with CloudSat retrievals allows to assess the influence of HIRHAM5 cloud and precipitation treatment. Mean snowfall rate for ERA-Interim is calculated from the monthly accumulation from the twice-daily values. Note, that HIRHAM5 simulations were carried out

before the release of ERA5.

## 2.3 CloudSat observations

CloudSat is part of the NASA A-Train (Stephens et al., 2002) constellation in a sun-synchronous orbit with an inclination of 98.2°. Therefore, it provides nearly global coverage reaching 82.5° from South to North (Tanelli et al., 2008) by a 16 day repeat cycle. The onboard Cloud Profiling Radar (CPR) operates at 94 GHz providing observations of the vertical distribu-

tion of clouds and light precipitation with the vertical resolution (bin) of 240 m (Tanelli et al., 2008). Its footprint is 1.4 km across and 2.5 km along track. The minimum detectable reflectivity is dependent on, e.g. cloud cover, seasonal changes in temperature, surface type, and atmospheric attenuation, typically varying by $\sim$ 1 dB over the globe in the range from -30.9 to -29.9 dBZ (Tanelli et al., 2008). Due to the relatively high frequency, the CPR signal can suffer from attenuation of atmospheric gases, e.g. of water vapor and oxygen. In addition, significant attenuation is also caused by the hydrometeors, and the

measurements may be affected by the multiple scattering effects (Mace et al., 2007; Marchand et al., 2008; Battaglia et al., 2010)





Here, we use two CloudSat products, the measured radar reflectivity vertical profile 2B-GEOPROF (Marchand and Mace, 2018) and the snow profile product 2C-SNOW-PROFILE (Wood and L'Ecuyer, 2018). The version 5 (R05) is used for both products. The 2B-GEOPROF includes the observed reflectivity corrected with the MODIS (Moderate-Resolution Imaging

Spectroradiometer) cloud mask product (Ackerman et al., 1998). The measured reflectivity may be attenuated which is not compensated in the product itself. Hence, in model-to-observation comparison the attenuation due to atmospheric gases and hydrometeors is included in the reflectivity computations (Sect. 2.4). The 2C-SNOW-PROFILE product includes estimates of particle size distribution and snowfall rate retrieved from the observed radar reflectivity applying ancillary meteorological information of the ECMWF-AUX (Stephens et al., 2008) and *a priori* information of snow microphysical and scattering

properties (Wood and L'Ecuyer, 2018). In the 2C-SNOW-PROFILE product, the precipitation presence and phase at the surface are primarily examined from additional 2C-PRECIP-COLUMN - product (Haynes, 2018), and secondary defined from the 2B-GEOPROF near-surface reflectivity values with the cloud mask correction and temperatures from ECMWF-AUX, for each radar profile within the retrieval algorithm. The snowfall is indicated and a snowfall rate is retrieved if the assessed melted fraction of precipitation is lower than 10%.

The retrieval of snowfall rate ($S$) from the measured reflectivity ($Z$) is a significant source of uncertainty when applying radar measurements to estimate snowfall rates, e.g. Bennartz et al. (2019). Hence, in the 2C-SNOW-PROFILE product, the snow retrieval is not based on a single pair parameter values of $Z$-$S$ relation, but is optimized by minimizing a cost function which represents differences between simulated and observed reflectivities and also differences between estimated and *a priori* values of the snow microphysical properties (Rodgers, 2000; Wood and L'Ecuyer, 2018; Edel et al., 2020). Milani et al. (2018)

found that adoptable $Z$-$S$ parametrizations considering the local microphysical conditions provide better performance than a method with static $Z$-$S$ relationship. Thus, we are confident to use the output of 2C-SNOW-PROFILE product as the ground truth, though acknowledging the relevant unreliability stemming from the uncertainties in observed reflectivities, the used retrieval parameters and its *a priori* assumptions (Edel et al., 2020).

As mentioned in the Sect. 1, due to clutter contamination, the CPR cannot reliably measure reflectivity near the surface

resulting in the blind zone. The magnitude and vertical extent of the enhanced reflectivity values related to back-scattered power from the surfaces vary depending on surface characteristics such as topography, roughness, and material (Palerme et al., 2019). In the 2C-SNOW-PROFILE product, the blind zone is determined as the two (four) bins above the bin containing the surface over the ocean (land) and the next highest bin, i.e. at altitude of $\sim$ 750 m (1200 m) over ocean (land) is considered for the snowfall retrieval (Wood and L'Ecuyer, 2018). Therefore, shallow precipitation or evaporation below this height might

lead to a deviation from the true near-surface snowfall rate. The resulting underestimation (sometimes also overestimation) of snowfall rate due to a 1000 m blind zone has been found to be rather small for Svalbard but higher at Belgian Princess Elisabeth station in East Antarctica (Maahn et al., 2014).

One of the important updates in the product version R05 is the use of an improved DEM (Digital Elevation Model) for the estimation of the surface height, and this should affect especially Greenland, which has steeply-varying terrain. With R05, the

number of observations suspected to be contaminated by ground clutter should have decreased (Palerme et al., 2019). When applying the 2C-SNOW-PROFILE product, we have determined the surface snowfall rate from the snow profile data utilizing



quality flagging of the snow retrieval status, following the example shown in (Palerme et al., 2019; Edel et al., 2020). The retrieval status is represented for each profile by an 8-bit array and an activated bit provides information about the retrieval performance (Wood and L'Ecuyer, 2018). We have utilized only those profiles, which have the zeroth and first bit field activated indicating that a snow layer is detected in the profile and snow is indicated at the surface. Additionally, if the third bit field is activated, meaning a large vertical gradient in snowfall rate between the near-surface bin and the bin immediately above is seen, the surface snowfall rate is determined from the second-lowest bin instead of the lowest near-surface bin. Such a strong gradient can be caused by surface clutter or by the presence of shallow precipitation. In the latter case, we are underestimating the surface snowfall rate. However, the other two possible reasons, the mentioned ground clutter contamination or the partial melting, can produce a significant error to the estimated surface snowfall rate. Palerme et al. (2019) pointed out that the third bit field was mainly activated on the edges of the fjords over the east coast of Greenland, and on the peaks of the Prince Charles Mountains expected to have clutter issues. Although also orographic precipitation can produce large snowfall gradients, however, in this case these flagged observations should cover larger areas along the ridges and coastline which were not observed by Palerme et al. (2019).

## 2.4 PAMTRA

The Passive and Active Microwave radiative TRAnsfer tool (PAMTRA; Mech et al., 2020) is a model framework to forward simulate passive and active microwave radiation through the cloudy atmosphere for up- and downward looking geometries. It can calculate polarized brightness temperatures and the full radar Doppler spectrum and its moments. PAMTRA requires input that describes the atmospheric state including hydrometeor contents and characteristics, instrument specifications, and the observation geometry.

In this study, the atmospheric state from HIRHAM5 was used to calculate the two-way gaseous attenuation of the radar beam using the gas absorption model by Rosenkranz (2015) including modifications of the water vapor continuum absorption (Turner et al., 2009) and the line width modification of the 22.235 GHz $H_2O$ line (Liljegren et al., 2005). To calculate the absorption/emission and scattering properties of hydrometeors the hydrometeor mixing ratios have been converted to particle size distributions following the microphysical assumptions of HIRHAM5 (Sect. A in Appendix). For each size, the back-scattering and extinction cross-sections are calculated and used for simulation of the radar reflectivity. For cloud liquid and rain particles, which can be assumed to be spherical, the Lorentz-Mie method is used and the refractive index of water is defined according to Turner et al. (2016). Cloud ice and snow particles have more complex structures than droplets or raindrops and the spheroidal or spherical approximations does not provide realistic scattering characteristics at 94 GHz (Tyynelä et al., 2011). Hence, for these particles, the Self-Similar Rayleigh Gans–Approximation (SSRGA; Hogan and Westbrook, 2014) is utilized and the coefficients needed to describe the ice particle properties for the scattering computations are derived as in Hogan et al. (2017). The refractive index of ice is taken from Mätzler (2006). From extinction and backscatter cross-sections both the attenuated (by gases and hydrometeors) and the unattenuated reflectivity have been calculated.

Multiple scattering may affect the observations (Battaglia et al., 2010) at 94 GHz and can be approximately 1 dB in snowfall with reflectivity values greater 10-15 dBZ (Matrosov and Battaglia, 2009), although not considered in these computations.



PAMTRA includes a set of different options to describe particle size distributions from mono-disperse, several functions, and fully resolved distributions. To be consistent with the microphysical scheme of the atmospheric model or the in situ measurements that provides the input, PAMTRA implements the same assumptions for the particle size distributions and particle properties (Sect. A).

## 245  3  Methodology

### 3.1  Sampling

HIRHAM5 model grid points and CloudSat CPR observations differ in space and time. These sampling differences need to be considered by spatial and temporal re-sampling in order to guarantee a fair comparison. The used sampling grid is an equal $1°$ $\times\ 1°$ grid. As stated in Souverijns et al. (2018), the CloudSat overpass occurring every couple of days is not representative to
describe individual snowstorm variability in a certain specific location. However with a large enough sampling grid CloudSat can on average produce a reliable climatology.

Within the observation-to-model approach for each sampling grid point, one daily value is retained taken as a mean of all the values of the CloudSat overpasses over the grid area and similarly for the HIRHAM5 modeled values. In case there are no CloudSat observations in the specific grid point, the daily value is excluded from the four-year analysis also from the model.
Because of the CloudSat orbit different grid-points are observed at different preferable times of day which is investigated in the model-to-observation approach (see below).

Because the analysis is performed on the $1°\ \times\ 1°$ grid, the number of model grid points per sampling grid cell decreases with latitudes due to the meridian convergence (Fig. 1). Due to its orbit the number of CloudSat measurements increases with latitude ((Edel et al., 2020); there in Fig. 1). Therefore, for both data sets the frequency of occurrence per grid point is calculated
and taken into account within statistical intercomparisons. When constructing joint histograms of temperature and reflectivity, so-called Contoured Frequency by Temperature Diagrams (CFTDs), the normalization is performed firstly by the number of samples in each grid point and secondly by the total sum of hits. In addition, the temporal sampling difference is considered by using the respective model output for each region closely coinciding with the times of the CloudSat overpasses. The stated model times for each region are shown in Table 1.
For the model-to-observation intercomparison we also need to consider the different vertical sampling, i.e. equal spacing by CloudSat and vertically stretched grid by HIRHAM5. Here, we defined the height bins to have a 250 m size below 1 km, a 500 m size between 1 and 4 km and a 750 m size between 4.0 and 10 km. The temperature bins are equally sized (2 K) between -70 and -10°C. In our analysis, we have restricted the temperatures to be below -10°C to exclude any effect of melting and melting layer from the statistical analyses of modeled and observed reflectivities. The temperature values for the PAMTRA modeled
reflectivities are taken from HIRHAM5 itself, and for the CloudSat observations, temperatures are obtained from ECMWF-AUX (Miller and Stephens, 2001). The sensitivity of the CPR is approximately - 28 dBZ, so the binning for reflectivities is carried out with 2 dBZ between -28.0 and 20 dBZ in the comparisons between modeled and measured reflectivity values.





However, model-only analysis includes wider range of values to demonstrate the small reflectivity values produced by cloud ice from the model, more discussion in the results (Sect. 5.2).

## 3.2 Circulation Weather Type classification

In order to better identify reasons for potential deviations between observations and HIRHAM5, we also composite snowfall maps for different distinguishable weather regimes and evaluate the model output to observations in each regime separately as performed in Akkermans et al. (2012). Here, we selected two sub-regions of the northern North Atlantic around Svalbard (Fig. 3d) covering the latitude band of 70°N - 81°N. The regions East (40°E - 10°E) and West (20°W - 10°E) are considered

such that the East regions is directly north of Scandinavia and includes Svalbard while the West region avoids land regions and is placed between Greenland and Svalbard. Both areas are characterized by high synoptic variability with frequent cyclone passages. The regime classification was performed with ERA-Interim 6-hourly 850 hPa geopotential height and shear vorticity for the studied period with the methodology of Jenkinson-Collison (Jenkinson and Collison, 1977; Philipp et al., 2016). Eight exclusionary directional classes according to compass points (N, NE, E, SE, S, SW, W, and NW) and two vorticity

circulation regimes (cyclonic (C) and anticyclonic (AC)) are specified, and the occurrence of each regime for both sub-regions are determined. Clearly, the northerly, southerly, and both vorticity classes are by far most frequent with an occurrence of 16% (25%), 12% (10%), 31% (38%) and 34% (32%), respectively for the East (West) sub-region. To simplify the analysis, the less frequent NE (6% (7%)) and NW (9% (6%)) were added to the northern regime typically representing situations when cold Arctic air masses move southward. Similarly, SW (7% (10%)) and SE (8% (3%)) were added to the southern cluster which is

a typical situation for warm air intrusions into the Arctic. Finally, we divided the modeled and observed daily mean snowfall rates to these four regimes and calculated the contribution to the yearly mean snowfall rate to each regime separately.

# 4 Results of observation-to-model evaluation

## 4.1 Comparison of modeled and retrieved surface snowfall rates

Two distinct regions of high (>500 mmyr$^{-1}$) average annual snowfall in the Arctic, namely the southeastern coast of Greenland

and the Atlantic storm track, are detected by both HIRHAM5 and CloudSat (Fig. 3). These are mostly related to cyclones which bring the heaviest snowfall during the snow accumulation season (September-May) in the regions of East Greenland, Barents, and Kara Seas, which are the dominant regions of the extreme cyclone occurrence. Typically 20 - 40 events per one winter season take place (Rinke et al., 2017). The cyclone activity is much stronger in the Arctic Atlantic than in the Pacific, i.e. cyclone snowfall accounts for approximately 80% of the total snowfall in these Atlantic regions, while cyclones account

for only circa 50% of total snowfall in the Pacific region (Webster et al., 2019). Additionally, lee cyclogenesis is important for precipitation production over southern and eastern Greenland (Rogers et al., 2004), whereas so-called Icelandic cyclones traveling further east are not favorable for precipitation over Greenland (Chen et al., 1997). This highlights the importance of the nudging in HIRHAM5 that leads to a good representation of cyclone associated snowfall as depicted in Fig. 2. The lowest





snowfall rates in both HIRHAM5 and Cloudsat are located in the Beaufort Sea, Canadian Archipelago, central of Greenland

Ice Sheet, and east Siberia with less than 100 mmyr$^{-1}$.

Across the whole Arctic, the area-weighted domain-average of the HIRHAM5 annual snowfall rate shows a nearly perfect agreement to the CloudSat based product with 213 (214) mmyr$^{-1}$ for HIRHAM5 (CloudSat). This value is also very similar to Edel et al. (2020) who derived 211 mmyr$^{-1}$ as CloudSat mean annual snowfall rate over the whole Arctic area when all profiles were included, but a rate of 183 mmyr$^{-1}$ when only profiles passing rigorous quality control were retained. Note that

the studied area is different between these two studies. CloudSat and HIRHAM5 agree much better among each other than they do with ERA-Interim which shows an annual area-weighted average Arctic snowfall rate of only 117 mmyr$^{-1}$ in our study area (Fig.A2).

When looking at the frequency distribution of the annual mean snowfall rates over all grid points, a rather similar distribution with highest occurrence of snowfall rates around 150 mmyr$^{-1}$ is apparent for HIRHAM5 and CloudSat (Fig. 4a). As a

minor difference HIRHAM5 shows more often rates between 100 - 300 mmyr$^{-1}$, whereas CloudSat has a tendency to higher extremes. The reasons for this could be the finer resolution of CloudSat resolving local precipitation hot spots or clutter contamination. In contrast to HIRHAM5 and CloudSat, the coarser scale ERA-Interim reanalysis has a much narrower frequency distribution and its distribution has a maximum at only 100 mmyr$^{-1}$. This demonstrates that although HIRHAM5 is driven by ERA-Interim, snowfall is determined by its physical parametrizations and these lead to an improved representation of snowfall.

We have proven that the chosen refined sampling has only minor effects on the results. Namely, the difference in mean yearly snowfall rate between the model values coinciding with CloudSat observations and model-only values (Fig. 3d) is on average over the whole Arctic region small. For 92% of grid points the difference is less than 20%. When the coincided sampling with CloudSat is applied, the modeled yearly snowfall rate is closer to CloudSat observations (the mean difference reduces from 2.2 mmyr$^{-1}$ to 0.7 mmyr$^{-1}$).

Given the overall good agreement, we now look at spatial differences in the annual snowfall rate across the full Arctic (Fig. 3c). Note, that the observations by CloudSat are generally in line with the results by Palerme et al. (2019) and Edel et al. (2020). Though model and observations show similar spatial distributions, distinct spatial differences occur. First, HIRHAM5 seems to consistently produce too high orographic precipitation than is detected by CloudSat over the coastal mountains, e.g. in Greenland, Norway, Svalbard, Novaya Zemlya, and in the Putorana plateau in Siberia. Therefore, this will also be investigated

via the model-to-observation approach (Sect. 5) which allows a closer look at the vertical structure. Second, while in many areas differences seem to be of random nature and can be attributed to the poor sampling by CloudSat, also two larger areas with systematic differences occur, namely an underestimation of HIRHAM5 along the North Atlantic storm track, Kara Sea, Baffin Bay, and Bering strait and an overestimation of HIRHAM5 over Greenland.

The North Atlantic storm track region sticks out as the largest area of a systematic underestimation of HIRHAM5 (Fig. 3c).

The underestimation is particularly strong southwest of Svalbard, the observed values are in between 500 - 1000 mmyr$^{-1}$ here. In contrast, the model provides values 400 - 700 mmyr$^{-1}$ across this region. However, when studying the uncertainty in the CloudSat climatology in more detail, Edel et al. (2020) identified the Arctic North Atlantic region as having relatively large uncertainty mainly due to a high frequency of possible mixed precipitation. In this respect it is important to look at the monthly





resolved snowfall distribution (Appendix Fig. A3 for HIRHAM5; Appendix Fig. A4 for differences). From September on, the region of highest snowfall in the North Atlantic moves more and more south with decreasing temperatures until its southern maximum extent in February/March. Interestingly, the highest model underestimation above 50 % (Fig. A4) over the Atlantic does not occur during the time of the strongest snowfall (January to March) but rather in the beginning of the snow season from September to November. This is reasonably in agreement with Akperov et al. (2018) who indicated an underestimated occurrence of deep cyclones in these months. Interestingly, this model underestimation seems to be coincident with the sea

ice-free areas of North Atlantic and also Baffin Bay as deduced from satellite data (Fig. A5) (Spreen et al., 2008). However, this is only a qualitative interpretation and requires more detailed examination in future studies. A clear statement whether and to what extent the HIRHAM5 underestimation in that region is related to model deficits or CloudSat uncertainty can not be given yet.

While for the region of southern Greenland a pronounced seasonal cycle in snowfall is evident, likely related to cyclone

activity, this is not the case for the northern Greenland region (Fig. 5). A noticeable feature is the strong overestimation of the modeled snowfall rates over the Greenland Ice Sheet and the coastal mountainous regions in the southern part as stated already in the yearly surface snowfall results (Fig. 3) (e.g. in the Greenland region with a median difference of 2.6 mm month$^{-1}$ for the model to show higher values calculated over the whole year). However, it should be noted that total precipitation in North Greenland is rather low and a high relative overestimation is not related to high snowfall rates. The annual distribution of the

snowfall is similar to the detailed snowfall climatology from CloudSat over the Greenland Ice Sheet shown in Bennartz et al. (2019), even though the magnitude is overestimated by the HIRHAM5 model.

In general, the seasonal course of snowfall rate for the different regions is well represented in HIRHAM5 as compared to CloudSat (Fig. 5). During the summer months, the model consistently overestimates the snowfall rates, however, the rates are also small in all regions, typically around 10 mm month$^{-1}$ or less, except in Greenland between 10 - 20 mm month$^{-1}$.

Again, the clear overestimation of the model is most visible during autumn, especially in the lower latitude band of the Kara Sea and Greenland regions. The North Atlantic (and the southern Chuckchi Sea) winter season stick out as the only where CloudSat shows higher snowfall rates than HIRHAM5. As discussed before here also retrieval problems related to mixed-phase precipitation might occur making it difficult to judge whether the model or the observations show deficits. Similar holds also for effects of the blind zone which therefore calls for an extended intercomparison in observation space.

The model's underestimation of the annual snowfall rate distribution in the North Atlantic and Kara Sea (Fig. 3c) is not visible in the region-wide averaged rates (Fig. 5), except in the winter season for the North Atlantic region. The reason lies in the averaging across the region. The model's overestimation over orographic and costal areas (e.g. over the Scandinavian Coast, Svalbard, and Novaya Zemlya) masks the model's underestimation over the oceanic regions. To expand our study, we zoom into two distinct regions in the North Atlantic corridor for defining the CWT regimes introduced in Sect. 3.2.

## 4.2 Circulation Weather Types (CWT)

Because the strongest underestimation of HIRHAM5 surface snowfall rate is seen in the North Atlantic, we selected two smaller sub-regions (depicted in Fig.3d) for the regime assessment with CWTs namely northerly (N), southerly (S), cyclonic (C), and





anticyclonic (AC) flow. The four different CWT regimes reveal consistently different snowfall distributions for both sub-regions (Fig.6; Fig. A6). Both the HIRHAM5-modeled and CloudSat-observed surface snowfall patterns agree well, which can
be explained by the nudging of HIRHAM5. Region-wide snowfall is brought in both sub-regions by cyclones which transport heat and moisture into the Arctic and accordingly the C regime brings most of the snowfall for the region.

In the Eastern sub-regions, during anticyclonic conditions (regime AC) a clear snowfall maximum northwest of Svalbard appears decreasing to the south and east both in HIRHAM5 and Cloudsat. With the southerly flow the highest snowfall accumulation appears in the northern part of the region generally above 76° N. During northerly flow, CloudSat shows that the majority
of snow falls in the region southeast of Svalbard (Fig. 6) which is sensible as this is a few hundred kilometers downwind of the ice edge and convection needs time to fully develop when the cold air flows over the relatively warm ocean. For this CWT regime, HIRHAM5 shows nearly a factor of two underestimation in the maximum snowfall rates in the southeast of Svalbard, also seen in the mean snowfall rates in Fig. 5. Similar characteristics can be found for the western sub-region (Fig. A6). As these both sub-regions relate to the area of largest underestimation southwest of Svalbard in annual snowfall rate (Fig. 3), the
poor representation of snowfall associated with northerly flow might be responsible for the overall HIRHAM5 underestimation. The northerly flow has a significant occurrence of 31%/38% for the East/West sub-region, and is often associated with Marine Cold Air Outbreaks (MCAO). MCAOs lead to organized convection when cold air flows over the relatively warm ocean, a phenomenon which many model struggle to represent (Geerts et al., 2021). The effect of underestimation during northerly flow might be partly compensated by cyclones which are associated with a higher snowfall rate in HIRHAM5 than by CloudSat.

## 5 Results of model-to-observation evaluation

Utilizing the PAMTRA forward-simulator, the HIRHAM5 model output of the mixing ratios of different hydrometeors can be converted to scattering properties, and the total simulated reflectivity can be compared to the one measured by CloudSat (2B-GEOPROF product, see Sect. 2.4) as described in Appendix A. This approach avoids assumptions in the snowfall rate retrieval from observations. Furthermore, it allows us to study the vertical structure of the hydrometeors particularly in respect
to orographic effects and the CloudSat blind zone. In this section, firstly we discuss the reflectivity distributions as a function of temperature for the different regions (Sect. 5.1) before we have a closer look at the the factors which determine the vertical reflectivity profile (Sect. 5.2) and investigate the performance of employing different cloud schemes within HIRHAM5 (Sect. 5.3).

### 5.1 Regional differences in reflectivity profiles

For investigating the differences in the vertical reflectivity structure between the different regions we focus on the winter season (DJF) which cover snowfall rates of approximately 30% over all seasons. Furthermore, we reduce problems related to mixed-phase conditions as temperatures are generally low. We follow Reitter et al. (2011) who build CFTDs instead of the often used geometrical height as it allows a better focus on the temperature-dependent cloud processes. CloudSat observations show the typical bi-modal structure in the CFTDs (Fig. 7) in nearly all regions with frequent occurrence of an ice cloud mode with





low reflectivities around -20 dBZ at low temperature of around -50°C and a second mode associated to snow with reflectivities
around 0 dBZ and temperatures warmer than -30°C. Note, that a minimum threshold of -15 dBZ is often utilized for identifying
snowfall in the 2C-SNOW-PROFILE product (Wood and L'Ecuyer, 2018; Haynes et al., 2009).

As the transition from ice clouds to snow is seamless a clear occurrence of maximum following a linear slope from low
reflectivities at cold temperature to high reflectivites at warm temperatures is found in CloudSat observations (red line depicted
in Fig. 7). This transition also reflects different snow growth processes, the depositional growth starting from around -50°C
and the dendritic growth zone at approximately -15°C leading typically through aggregation to enhanced snowfall rates. Also
different snowfall types such as shallow cumuliform and deeper nimbostratus snowfall events are associated with different
CFTDs as demonstrated by Kulie et al. (2016). Therefore, when averaging over larger regions and seasons this linear pattern
becomes dominant as for example in the global CloudSat CFTD by Reitter et al. (2011). Clearly this behaviour can not be seen
in the HIRHAM5 simulations. Hardly any reflectivities in regions colder than -35°C are produced indicating a problem with
ice clouds which will be addressed in more detail in the next subsection.

Due to the lower occurrence of cold temperature reflectivities, reflectivities at warmer temperatures are relatively more
frequent in HIRHAM5 than in CloudSat observation. However, HIRHAM5 is able to reproduce regional differences seen by
Cloudsat correctly. Enhanced reflectivity related to the snow mode (- 10 and 5 dBZ) occurs at the warmest temperature in the
North Atlantic (around -10°C) in both observations and model, similar at slightly warmer temperature in the Kara Sea regions.
In the Chukchi Sea, occurrences are confined to a narrow temperature range between -20 and -35°C, while in the Laptev Sea
the distribution broadens to colder temperature again in both observations and simulations. In the Chukchi Sea, HIRHAM5 can
also reproduce the increased reflectivity occurrence around -20°C in the lower latitude region compared to the higher latitude
region. The strongest difference between the observed and simulated CFTDs is visible for Greenland where the simulations
show reflectivities at much warmer temperatures and higher reflectivities consistent with the overestimation in snowfall rate by
HIRHAM5 discussed before.

We also checked the effect of attenuation in the simulations by comparing the attenuated reflectivity values with the non-
attenuated values. The attenuated CFTDs are 5.1% closer to the observed CFTDs, and generally, even with attenuation con-
sidered, the model sees higher reflectivities in warmer temperatures. Except in the case of Greenland, where the observed
occurrences of higher reflectivities are higher than the simulated ones. Additionally, the effect of using the modeled values
concurrent with the expected CloudSat overpasses (Fig. 1 and Table 1) in respect to all modeled values is examined. Typically
differences show a random geographical distribution in particular in regions where occurrences are low. The overall improve-
ment in agreement is 0.8 % for the observed reflectivity values when only concurrent model values are used. Thus, the exact
matching is considered insignificant except for times with little snow as in summer in some regions.

## 5.2 Vertical structure of hydrometeors

Because CFTDs average over different surfaces (ocean, sea ice, and land), we now investigate the vertical reflectivity structure
for a latitude band of 72°N - 73°N (circle indicated in Fig. 3c) again for the winter season. The Fig. 8 shows the seasonal
mean reflectivity cross section along the latitude circle together with the mean surface snowfall rate to emphasize the reasons





with strong snowfall in south-eastern Greenland and across the North Atlantic ocean. The importance of orography is clearly

evident. HIRHAM5 shows orographic effects with reflectivity enhancement reaching mid-tropospheric levels at both coasts of Greenland, over Baffin mountains and Novaya Zemlya. These structures are even more obvious in the differences to CloudSat, where HIRHAM5 shows strong overestimations (spikes) for nearly all grid point associated with strong orographic slopes. The strong vertical extent of these reflectivity signatures in HIRHAM5 is not visible in the observations and led us to conclude that this is a model deficit.

Looking at the CloudSat observations, the effect of ground clutter which causes a deeper blind zone over land (up to 1200 m) is most visible for Greenland and Baffin Island (with elevations up to 2000 m). Clutter filtering might cause the possible model overestimation over Greenland in winter (Fig. 5), which has also been found by (Edel et al., 2020) for the Arctic-wide average. Consistent with the CFTDs, reflectivities of more than -20 dBZ can be found in much higher (colder) parts of the atmosphere by CloudSat than in HIRHAM5 which will be investigated next.

**5.3    Differences between the different cloud microphysical schemes**

The simulated reflectivities can provide additional insight into the contributing portions of different hydrometeors to the total reflectivity. At first, we look at the HIRHAM5 control run employing the modified Tompkins scheme. Again for winter, we show the reflectivity by different hydrometeor types along the latitude belt in Fig. 9a. Snow particles clearly contribute the most to the simulated reflectivity for all heights and also throughout all the seasons. Even in summer (not shown), when the snow

particles manifest higher in altitude to the reflectivity, their contribution to reflectivity dominates over that of rain particles. The highest reflectivity values due to rain particles are concentrated in the North Atlantic region and some higher values are also modeled in the East Siberian Sea and the Beaufort Sea. Cloud ice produces reflectivities over the full troposphere, however, these are rather low in particular in the higher troposphere. This likely originates from the threshold of ice particle radius to be monodisperse of 40 $\mu$m (see Appendix. A), which results in very low reflectivity values. In contrast, the smallest radius of

particle in the snow class is 0.1 mm, and thus, snow always produces significant snow reflectivity.

Even in winter, cloud liquid water is present within the lowest four kilometer and produces significant reflectivities close to the surface over the North Atlantic. During the summer months, their contribution is more widely distributed to the studied latitude ring and higher in altitude (not shown), as to be excepted. The presence of low-level clouds and rain over the North Atlantic points at the difficulties to both (i) retrieve snowfall in mixed-phase conditions and (ii) to simulate reflectivity as

melting might produce complex particles which are not taken into account in the forward simulation.

In addition to the control run with the modified Tompkins scheme, two more runs with different cloud microphysical schemes, i.e. the original Tompkins and the Sundqvist schemes, are performed and their results are compared (Fig. 9b). For all schemes, the overall total mean reflectivity profile compared to the CloudSat observed reflectivity profile is close to similar. The mean reflectivity difference to CloudSat is varying between seasons and schemes, but generally the mean difference ranges

between 0.3 - 2.3% and no clear statement can be made, which of the schemes in general would be closest to reproduce the total reflectivity values compared to observations.





However, there are distinct dissimilarities between the schemes when the contributions to reflectivity by different hydrometeors are examined (Fig. A7). The original Tompkins scheme seems to produce more cloud liquid particles than the modified Tompkins or Sundqvist scheme. Similar is stated in Klaus et al. (2016), connected to the enabled more enhanced Bergeron-
Findeisen process, and thus associated with an increase of cloud ice particles. The finding of Klaus et al. (2016) is limited over sea ice areas only, and this is not obvious from our analysis here over a lower latitude band (72°N - 73°N) from these model runs. Another difference, is that the Sundqvist scheme produces rain particles, though small amounts, reaching high levels of atmosphere, which is not the case with either of the Tompkins schemes. The notable dissimilarity between the schemes is the contribution of cloud ice and snow particles to reflectivity. As both Tompkins schemes seem to have high fraction of snow par-
ticles especially in the lower altitudes, the Sundqvist scheme tends to have higher fractions of cloud ice particles contributing even in the lower altitudes. The above mentioned scheme differences seem to be seasonally independent, i.e. similar features can be seen not only during winter as shown in the Fig.A7. While we are not going into more detail here, it is clear that the model-to-observation approach allows to investigate the relative performance of different schemes. The next step would be to identify regimes, such as based on temperature that show especially high differences between the cloud schemes, and
subsequently compare observations and different model simulation for the these regimes.

## 6 Conclusions

This study investigates how well a regional climate model, in this case HIRHAM5, can represent the Arctic snowfall both regionally and seasonally, compared to the CloudSat retrieved surface snowfall rates and observed reflectivity profiles. We identified the specific weather types related surface snowfall rates, and their patterns over the northern North Atlantic.
Firstly, in the observation-to-model approach, the modeled surface snowfall rate is compared to the 2C-SNOW-PROFILE product at yearly and monthly scales. The average yearly modeled snowfall rate (213 mmyr$^{-1}$) agrees with the retrieved values (214 mmyr$^{-1}$), and the spatial distributions are similar. This includes that the patterns of the storm tracks related snowfall over the northern North Atlantic in winter and over the Baffin Bay seen as increased snowfall rates in the west coast of Greenland during autumn. The seasonality of snowfall rates over the Siberian seas (lower in winter, higher in summer) is also repre-
sented well. One of the clear differences is found for the magnitude of the orographic snowfall over the (coastal) mountains, e.g. in Greenland, Norway, Svalbard, Novaya Zemlya, and in the Putorana plateau in Siberia, where the model significantly overestimates the snowfall rates, compared to CloudSat. Another difference is the underestimation of the magnitude of the snowfall rate over the northern North Atlantic in the model, compared to CloudSat. This seems partly be caused by the poor representation of MCAOs in the simulations and by CloudSat uncertainty related with mixed-phase precipitation over the open
sea. Follow-up in-depth studies on this is required.

Secondly, in the model-to-observation approach, the model output is applied to the forward-simulator PAMTRA to compute the reflectivity, i.e. the simulated reflectivity is compared with the CloudSat CPR measured reflectivity (2B-GEOPROF product). The results support the surface snowfall rate findings. The model and observations show enhanced reflectivity over the storm track regions, especially over the northern North Atlantic during autumn and winter. The observations of the Greenland





surface layer and, e.g. the Baffin mountain range are clearly contaminated by the clutter. However, also the modeled overestimation of orographic precipitation is pronounced on the coasts. Generally, it seems that the modeled attenuation is higher than actually seen in the observations, especially during summer months. However, without considering attenuation in the simulations, the model overestimation increased by more than 10 - 20%.

Based on the CFTD analysis, the CPR observed reflectivity shows higher occurrences in the colder regime (i.e. at gener-
ally higher altitude), while the modeled occurrences dominate at the lower altitudes (warmer temperatures). In the model, the threshold of ice particle radius is 40 $\mu$m, and thus, the simulated reflectivity occurrences are often for the ice particles below the -30 dBZ threshold regime, and the snow growth process is inadequately depicted in the profile. The enhanced reflectivity during autumn and winter at temperatures close to -10°C (to interpret as close to the surface level) is seen both in the model and observations and indicates the frequent snowfall due to cyclone activity. Especially during autumn, but also seen
in winter months, the observations show a pronounce bi-modal feature possibly indicating different snow growth processes, the depositional and dendritic growths, and/or a presence of two different snowfall categories, the shallow cumuliform and the thicker nimbostratus snowfall, studied in Kulie et al. (2016). Further analysis is needed to interpret the ice growth processes and related simulated reflectivities, as these were clearly one of the significant differences between the model and observed values. Additionally, the other CloudSat retrieved products such as the classification of 2B-CLDCLASS product could be used
to investigate more detailed the representation of the microphysical processes in the model.





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



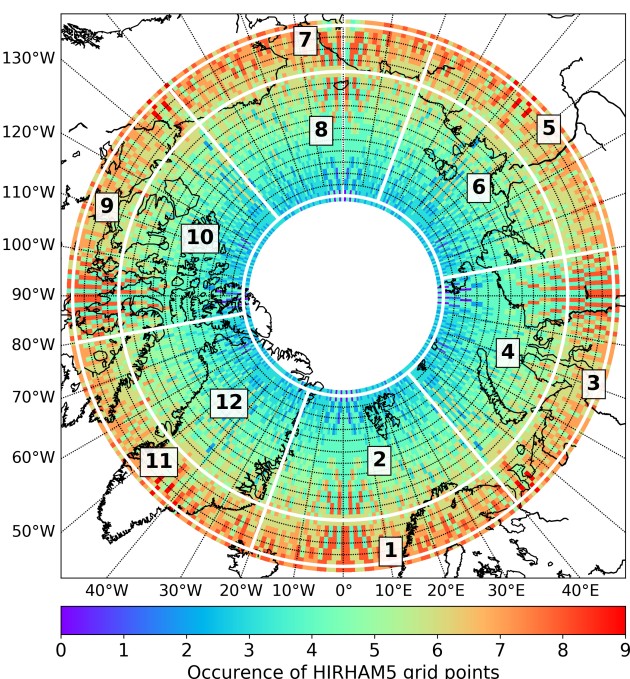

**Figure 1.** The studied regions (described in Table 1) shown on map with the occurrence of HIRHAM5 model grid points in the $1° \times 1°$ sampling grid points.

**Figures**



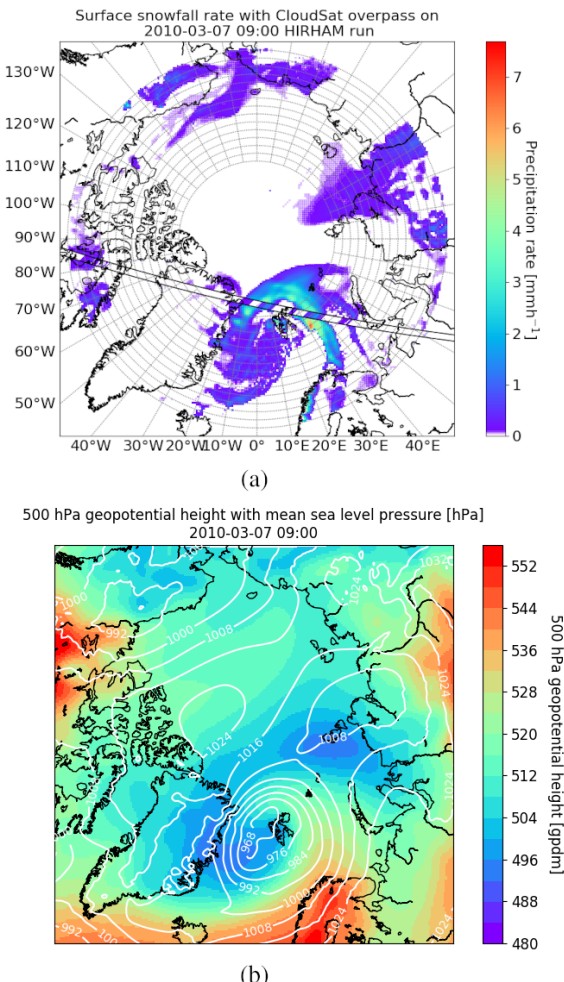

(a)

(b)

**Figure 2.** Case study on March 7, 2010. (a) The overpass of CloudSat, where the surface snowfall rate (in mmh$^{-1}$) taken from the closest model run of HIRHAM5 is shown underneath and the overpass is colored with surface snowfall rate values of 2C-SNOW-PROFILE-product. b) shows HIRHAM5 500 hPa geopotential height with mean sea level pressure plotted with contour lines.



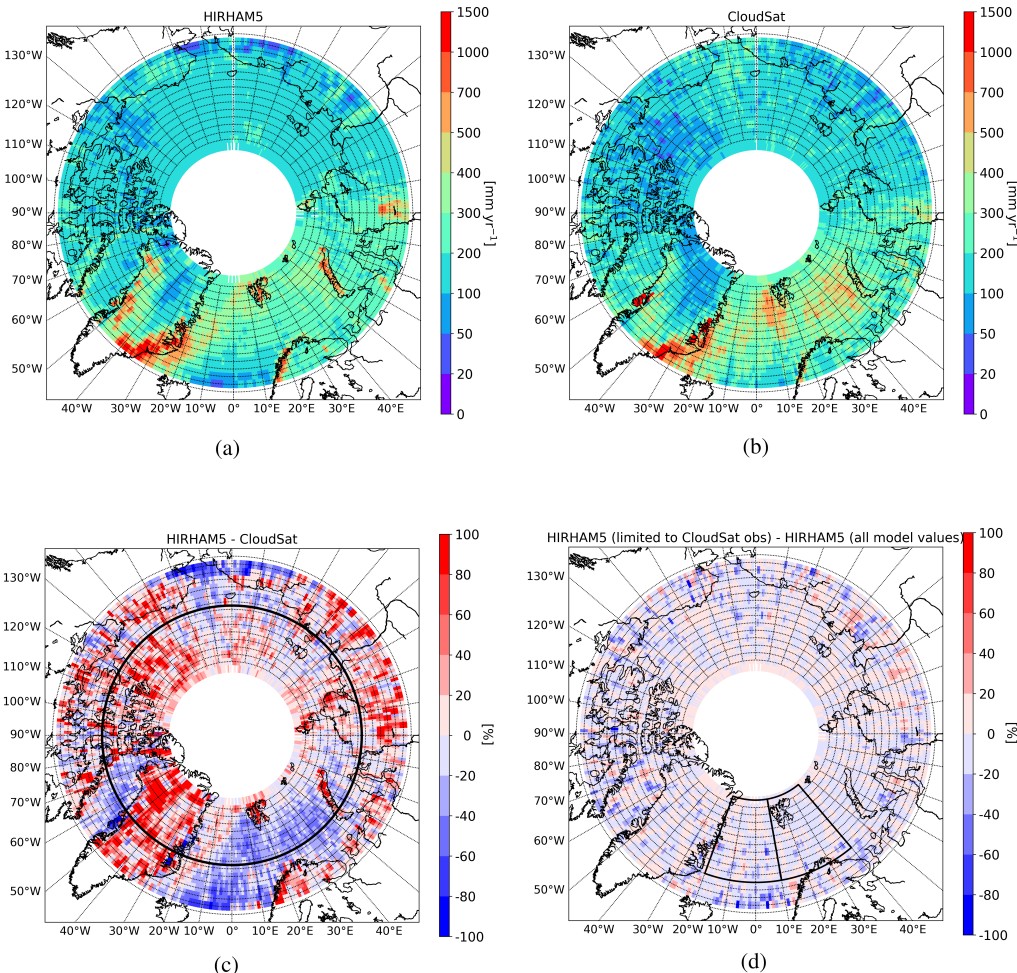

**Figure 3.** Annual mean snowfall rates during 2007-2010. a) modeled with HIRHAM5 when periods of CloudSat overpasses are considered, b) observed with CloudSat and retrieved with 2C-SNOW-PROFILE product, c) the difference of snowfall rates between HIRHAM5 and CloudSat, and d) the difference of the modeled snowfall rates with HIRHAM5, with and without considering the times of CloudSat overpasses. In c) difference plot, the warm colors show that HIRHAM5 has higher rates, whereas colder colors indicate that CloudSat provides higher rates. Percentages are calculated related to Cloudsat observations. The thicker black line in c) show the latitude band of 72°-73° N. In d) difference plot, the warm colors indicate that considering the times of CloudSat overpasses when calculating the mean shows higher rates and cold colors that when using all daily modeled values the mean is higher. Percentages are calculated related to model-only values. The sections marked in d) show the two sub-regions in the northern North Atlantic for the CWT analysis.





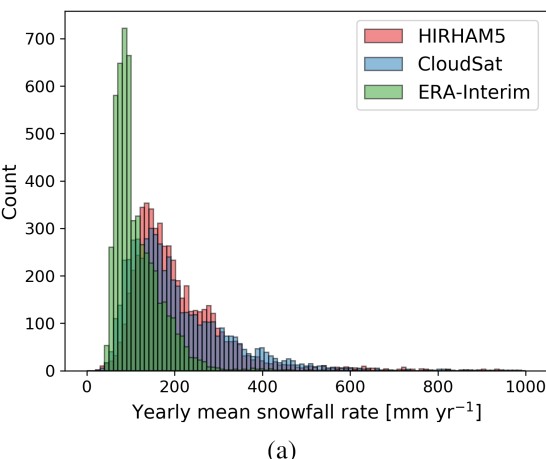

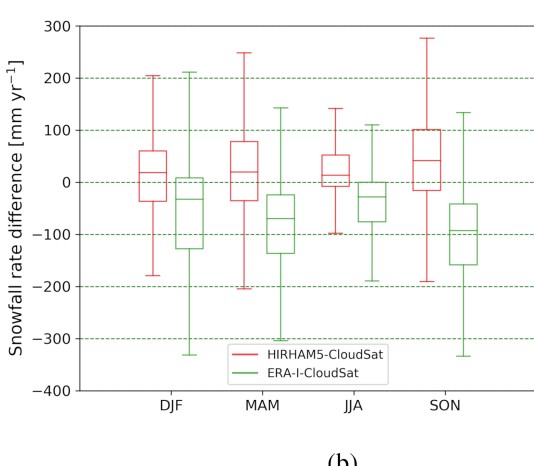

(a)  (b)

**Figure 4.** a) Histogram of yearly mean snowfall rates of HIRHAM5, CloudSat, and ERA-Interim and b) boxplot of the difference in yearly snowfall rate between HIRHAM5 and CloudSat (red) and ERA-Interim and CloudSat (green) distributed seasonally. The box extends from the lower to upper quartile values, the line shows the median, and the whiskers show the range of the difference values as the first and third quartiles.





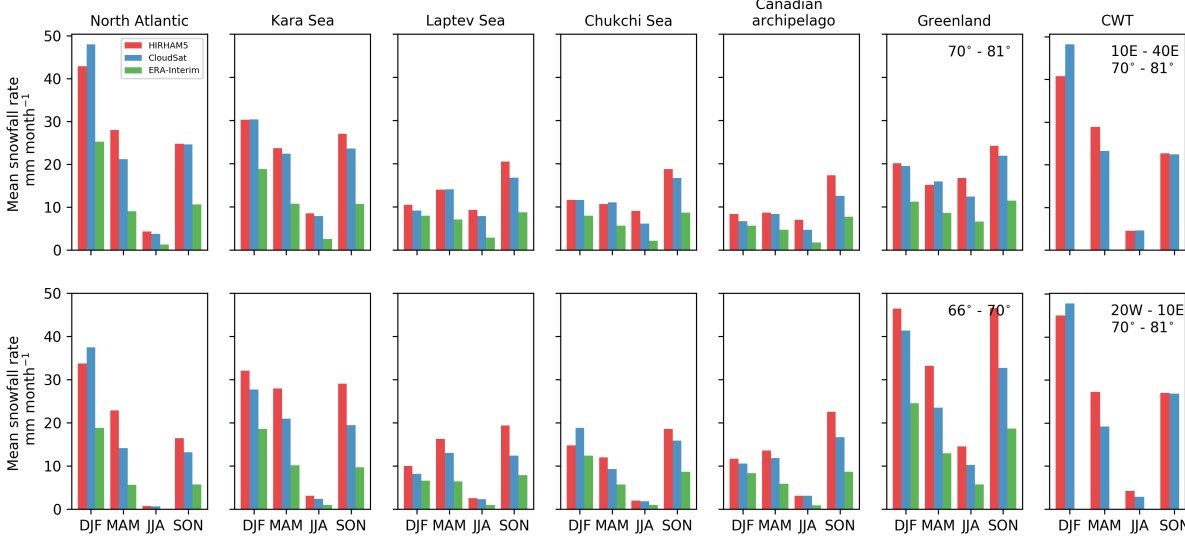

**Figure 5.** Monthly area-weighted mean snowfall rates of HIRHAM5, CloudSat and ERA-Interim shown seasonally for each studied region.The latitude band 70° N-81° N shown in the upper and the band 66° N-70° N in the lower panel. In the rightmost column, monthly area-weighted mean snowfall rates of HIRHAM5 and CloudSat are shown for the both North Atlantic CWT regions.

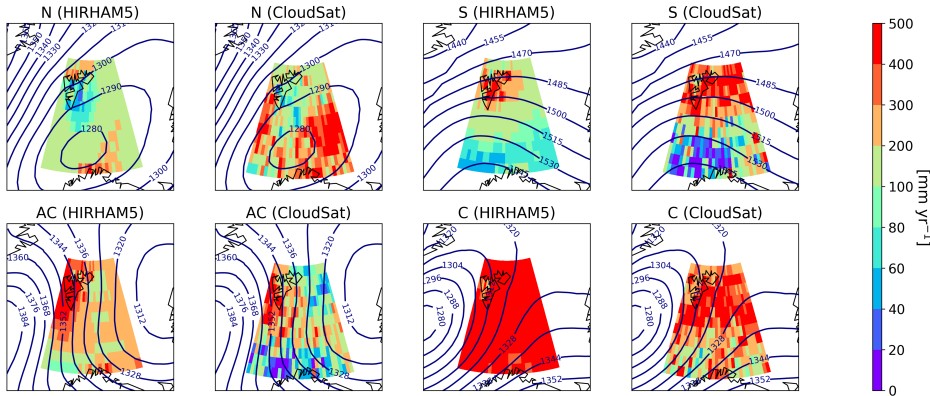

**Figure 6.** Snowfall rate composited to different CWTs, N (including N, NE, and NW) S (S, SE, and SW), C and AC, for the area of 10° E-40° E (the colored sub-region is according Fig. 3d) between latitude bands 70°- 81°N for both HIRHAM5 and CloudSat. The mean 850 hPa geopotential height (gpdm) associated with the CWTs is also shown as blue contour lines.







**Figure 7.** The observed and modeled CFTD for the both latitude band regions on winter season (DJF, 2007-2010). On the x-axis is the reflectivity in dBZ, on the y-axis is the temperature from -70°C to -10°C, normalization is done by the sum of total hits, which varies from region and season, but is ranging between 86500 - 6.6$\dot{1}0^6$.



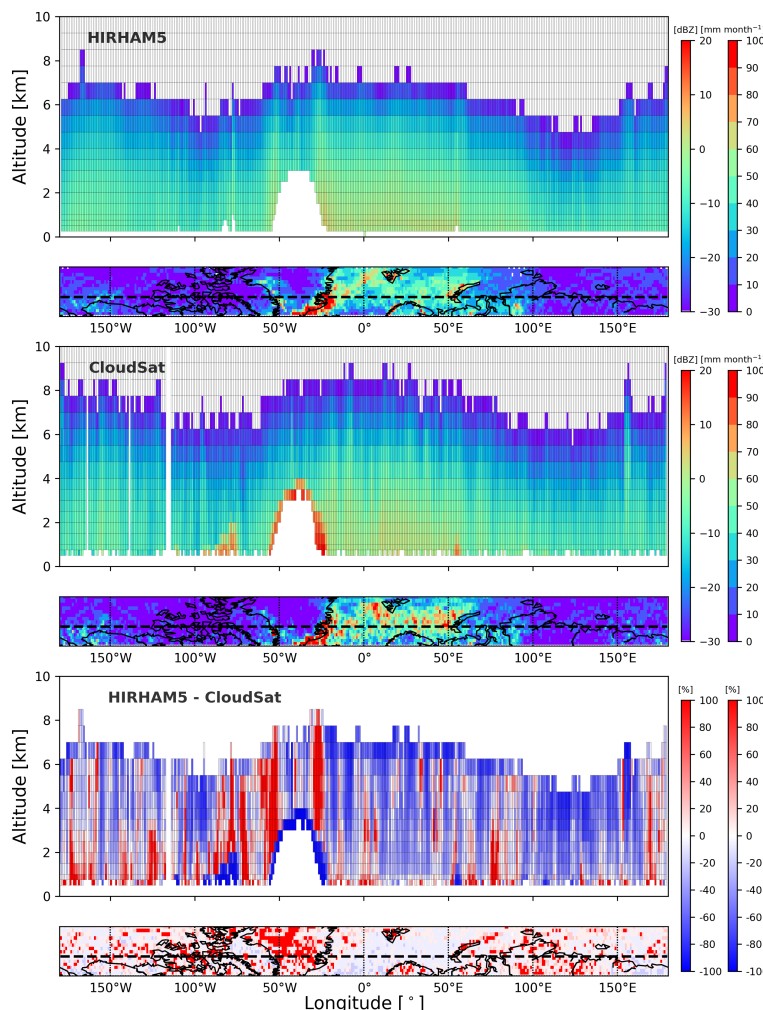

**Figure 8.** Mean modeled reflectivity of all hydrometers of HIRHAM5 (top), observed by CloudSat (middle) and the difference of values (bottom) as a function of altitude for a latitude band of 72° N-73° N during winter (DJF). In the difference plot, the red color indicates that HIRHAM5 simulates a higher reflectivity than CloudSat observations, and blue colors vice versa. The small boxes below each of the reflectivity profiles show the monthly surface snowfall rates and the difference. The shown altitude is restricted to higher than 500 m and the difference in percents is calculated subtracting in linear scale the CPR measured value from the modeled one, divided by the measured one, and multiplied by 100.



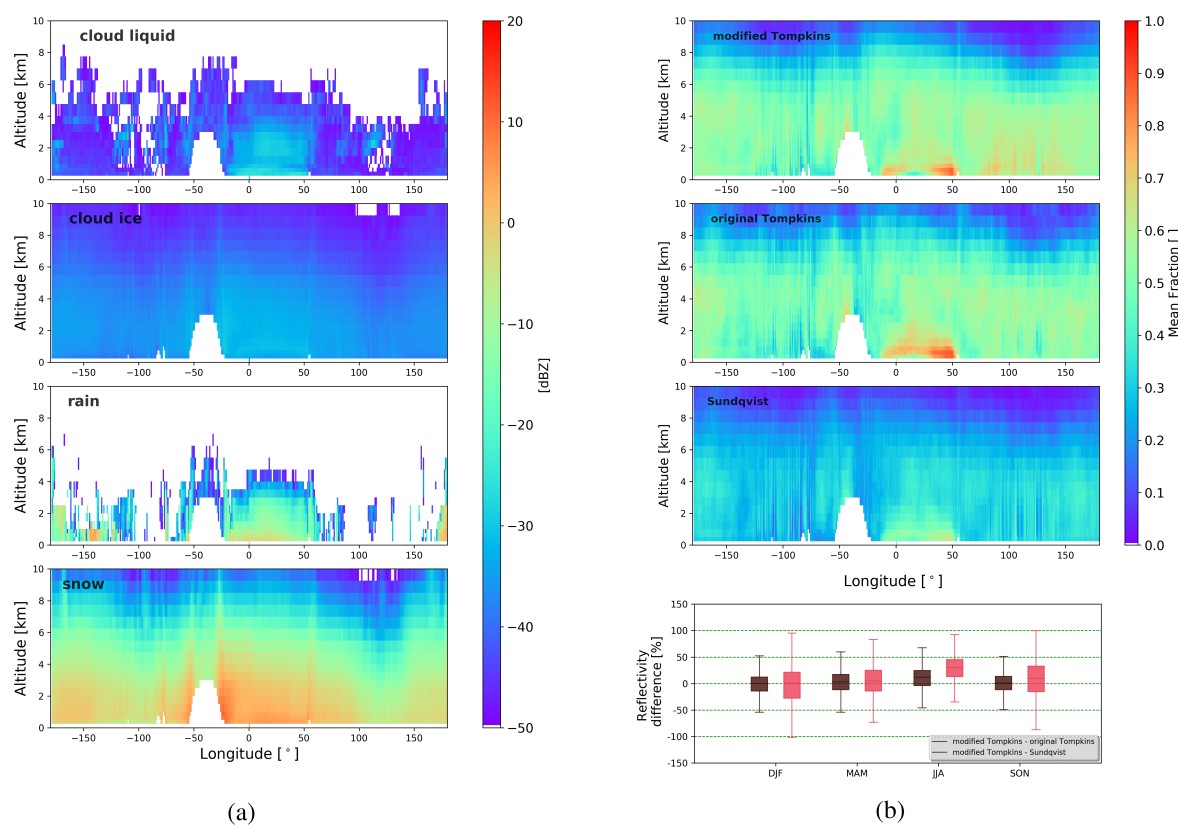

(a)                                    (b)

**Figure 9.** a) Mean modeled reflectivity with modified Tompkins scheme separately for all hydrometers (cloud liquid and ice, snow and rain) as a function of altitude for a latitude band 72°-73° N for the winter season (DJF), b) mean fraction of reflectivity contributed by snow with the three different microphysical schemes in the winter season, and c) box plot of the reflectivity differences between the modified Tompkins scheme in respect to the two other schemes (difference to the original Tompkins with brown and to the Sundqvist with rosa) seasonally for snow.





## Appendix A: PAMTRA simulations based on HIRHAM5 model output

For each grid point and model level, the HIRHAM5 provides the mass mixing ratios of all hydrometeor classes: cloud ice
and droplets, snow and rain particles. From the ratios, the particle size distributions (PSDs) are calculated as well as particle
properties such as size, shape, and density, following the ECHAM5 microphysical assumptions and are used as input into
PAMTRA.

In HIRHAM5, all particles are basically assumed to be spherical, and described with constant densities; for cloud droplets
and rain particles of $\rho_w = 1000 \, \text{kgm}^{-3}$, for ice particle $\rho_i = 500 \, \text{kgm}^{-3}$, and for snow particles $\rho_s = 100 \, \text{kgm}^{-3}$. The reference
density of air is $\rho_0 = 1.29 \, \text{kgm}^{-3}$ and the density of dry air $\rho_a$ is calculated at the corresponding height.

HIRHAM5 provides the mass mixing ratios $q_l$ and $q_i$ as prognostic variables for the cloud liquid and ice, respectively. The
mean volume radius of cloud droplets can be obtained as in Roeckner et al. (2003) by

$$r_{vl} = \left[ \frac{3\rho_a q_l}{4\pi N_l \rho_w} \right]^{1/3},$$ (A1)

where the cloud droplet distribution $N_l$ is assumed to have mono-disperse distribution declining exponentially between fixed
values of lower $N_{llt}$ and upper troposphere $N_{lut}$ (Roeckner et al., 2003). The $N_l$ can be described as a function of pressure $p$
in Pa (Stevens et al., 2013)

$$N_l = N_{lut} + (N_{llt} - N_{lut}) \exp\left[ 1 - \left( \frac{p_{lt}}{\max(10000, p)} \right)^2 \right],$$ (A2)

where the pressure top value of $p_{lt} = 80000$ Pa for lower troposphere is assumed. The fixed values for number concentrations
depend on whether the air mass column is over land $N_{llt} = 220 \, \text{cm}^{-3}$ and over ocean-ice $N_{llt} = 80 \, \text{cm}^{-3}$ and in the upper
troposphere the value decreases to $N_{lut} = 50 \, \text{cm}^{-3}$ irrespective of surface type.

For cloud ice, the effective radius $r_{ei}$ is determined with mean volume radius $r_{vi}$ in meters (Levkov et al., 1992; Roeckner
et al., 2003) defined as:

$$r_{vi}[m] = 10^{-6}(\sqrt{2809r_{ei}^3 + 5113188} - 2261)^{1/3},$$ (A3)

and ice particle distribution $N_i$ is also assumed as mono disperse (Potter, 1991)

$$N_i = \frac{6\rho_a q_i}{\pi \rho_i D_{vi}{}^3},$$ (A4)

where $D_{vi}$ is the mean volume diameter of ice particles.

For the precipitation, both snowfall and rain, HIRHAM5 output provides the fluxes divided into two components, large-scale
$F_{ls,snow/rain}$ and convective $F_{c,snow/rain}$. The convective component was assumed to be small (constituting approximately
less than 1% in precipitation rate and the occurrence frequency an order of magnitude smaller) compared to large-scale, and
therefore it is not considered in the forward simulations. The large-scale fluxes are converted to mass mixing ratios. For snow
mixing ratio

$$\rho_a q_s = \left( \frac{F_{ls,snow}}{C_{pr}a_{11}} \right)^{\frac{1}{1+b_{10}}},$$ (A5)





where fractional cloud cover $C_{pr} = 1$ is assumed and the velocity-dimensional parametrization of Heymsfield and Donner (1990), with $a_{11} = 3.29$ and $b_{10} = 0.16$, are employed. Hence, the slope of snow particle size distribution can be determined as in Potter (1991)

$$\lambda_s = \frac{\pi \rho_s n_{0s}}{\rho_a q_s}^{1/4},\tag{A6}$$

where exponential snow particle distribution $N(D_s) = n_{0s} \exp(-\lambda_s D_s)$ of Gunn and Marshall (1958) is followed with a constant intercept parameter $n_{0s} = 3 \cdot 10^6$ m$^{-4}$. For the snow class, a minimum ice crystal size is set to $r_{s0} = 10^{-4}$ m.

The rain drop size size distribution is following the Marshall-Palmer distribution $N(D_r) = n_{0r} \exp(-\lambda_r D_r)$ with value of $n_{0r} = 8 \cdot 10^6$ m$^{-4}$ (Marshall and Palmer, 1948) and the mass mixing ratio $q_r$ and slope parameter $\lambda_r$ can be defined as follows

$$\rho_a q_r = \frac{F_{ls,rain}}{C_{pr} a_{10} (n_{0r})^{-1/8} \sqrt{\rho_0/\rho_a}}^{8/9}\tag{A7}$$

and

$$\lambda_r = \frac{\pi \rho_w n_{0r}}{\rho_a q_r}^{1/4},\tag{A8}$$

where fall velocity of rain drops is parametrized according to Kessler (1969)

$$v_r = a_{10}\{\frac{\rho_a q_r}{n_{0r}}\}^{1/8}\{\frac{\rho_0}{\rho_a}\}^{1/2},\tag{A9}$$

with $C_{pr} = 1$ and $a_{10} = 90.8$.

## Appendix B: Additional Figures of evaluating HIRHAM5 in respect to CloudSat retrievals and observations

This annex section includes the additional Figures to demonstrate the wider picture of the analysis in comparing the outputs HIRHAM5 with the CloudSat snowfall retrieval results and reflectivity observations. Figures are referred in the main text.

An example, a simulated radar reflectivity cross section is shown in Fig. A1a together with its observational counterpart from for a satellite overpass on March 7, 2010. The surface clutter is well distinguished in the CloudSat observations, especially over Greenland. The cut-off threshold of -28 dBZ following the sensitivity of CPR is applied for both simulations and observations. Generally, the agreement in vertical structure of precipitation is good. Due to the lower spatial resolution of the model, the simulated clouds are more widespread than observed. However, differences in the magnitude and cloud top height are evident.

In Fig. A1b, a comparison of the snowfall retrieval results calculated with PAMTRA simulator are compared to the known retrieval values from the literature for one representative profile during a precipitating event. This is performed more or less as a sanity check for the scattering assumptions in PAMTRA, and how close the scattering computations are to the values reported in the literature (Table A1). Though the simulated reflectivity is slightly underestimated over the full profile, the larges differences occur in the upper troposphere which is likely due to cloud ice. There is hardly any cloud ice above 6000 m in HIRHAM5, and thus cloud height seems to be lower for the simulated reflectivity than for the observed. Actually, the





HIRHAM5 model has an intrinsic upper threshold for the size of cloud ice particles with a diameter of 40 $\mu$m. With this size, the calculated back-scattering cross-sections for ice particles result in reflectivity values below the cut-off threshold defined by CPR.

840    In HIRHAM5, the reanalysis ERA Interim is used with the grid point nudging to control the simulated large-scale flow, however, as the modeled snowfall rate values significantly deviate between these models, the ECHAM5 microphysical parameterization seem to play a critical role in modeling the Arctic snowfall. The Fig. A2 shows the annual mean snowfall rate of ERA-Interim and the difference of snowfall rates between ERA-Interim and CloudSat. The underestimation of the ERA-Interim is very clear, although the similar patterns such as the influence of the cyclone track in the Arctic North Atlantic region

845  is accordingly modeled in the reanalyses.





**Table A1.** The snowfall retrieval relations used in calculating the example profile in Fig. A1.

| $Z_e(S)$ | Reference |
| --- | --- |
| $Z_e = 13.16 S^{1.40}$ | Kulie and Bennartz (2009) |
| $Z_e = 56.43 S^{1.52}$ | |
| $Z_e = 2.19 S^{1.40}$ | |
| $Z_e = 11.50 S^{1.25}$ | Liu (2008) |
| $Z_e = 10.00 S^{0.80}$ | Matrosov (2007) |
| $Z_e = S^{1.52}$ | Heymsfield et al. (2016) |

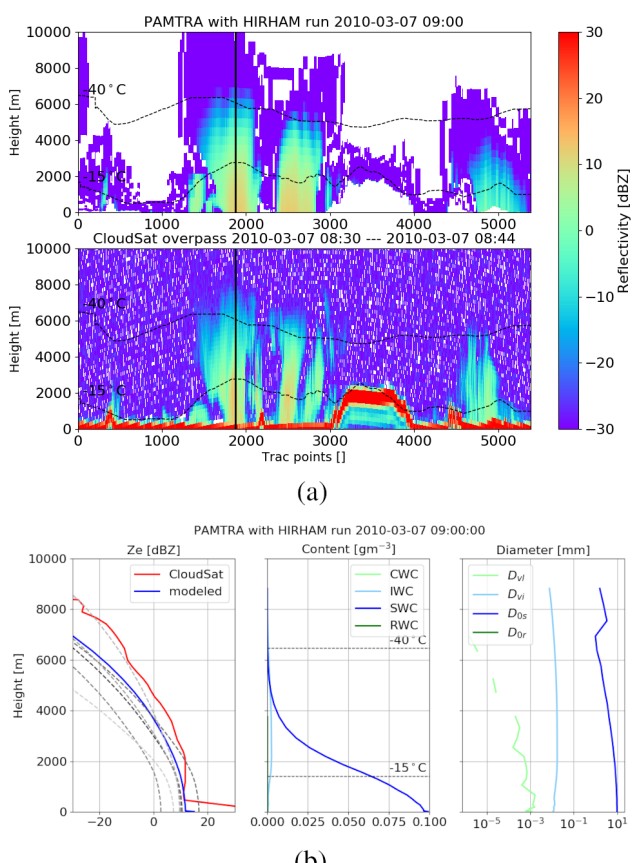

**Figure A1.** Case study on March 7, 2010. In a), above is a reflectivity simulated by PAMTRA with the mixing ratios produced by the HIRHAM5 and below is the observed reflectivity of CloudSat. In b) is shown for the profile location marked as a black solid line in a); on the left are the profiles of reflectivity with both CloudSat (red line), modeled (blue line) and example values from literature (stated in the supplement material Table A1) with corresponding snow mixing ratio (gray dashed lines), in the middle are cloud liquid (light green) and ice (light blue), rain water (green) and snow equivalent liquid (blue) contents, and on the right, are the different mean diameter profiles for the hydrometeors i.e. mean volume diameter of cloud liquid (light green) and ice (light blue), median volume diameter of rain (green) and snow particles (blue).





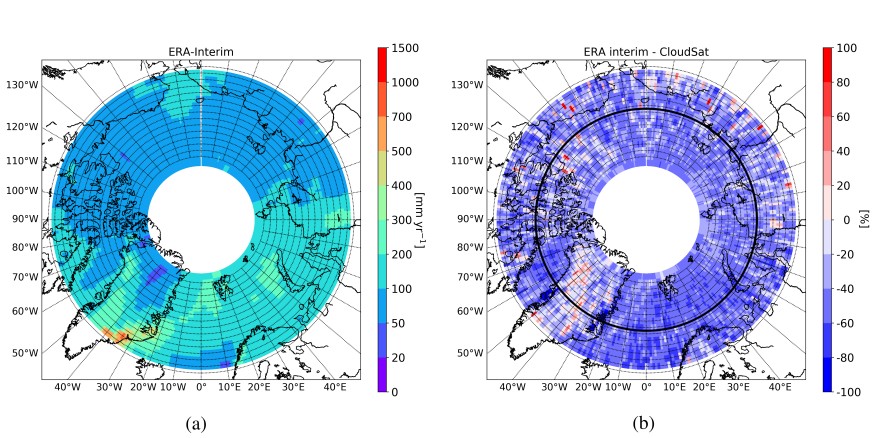

**Figure A2.** a) Yearly mean snowfall rate of ERA-Interim. b) Difference of snowfall rates between ERA-Interim and CloudSat.





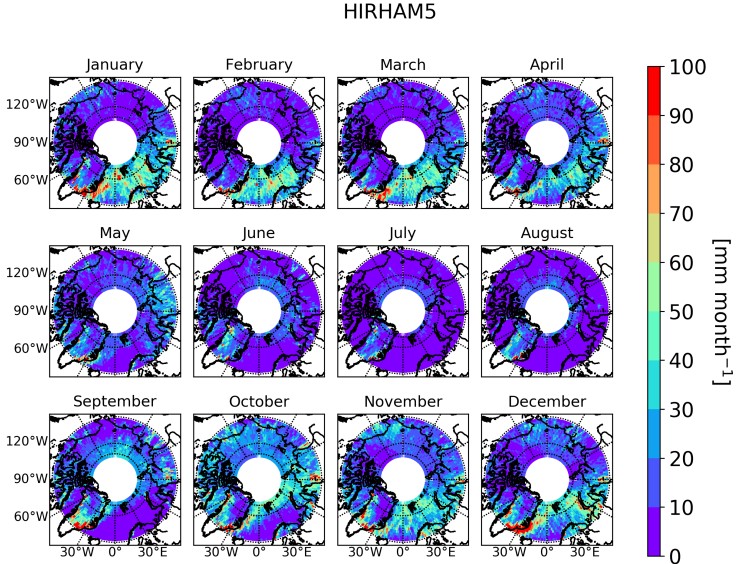

**Figure A3.** Monthly mean snowfall rate of HIRHAM5 when using the values coinciding with CloudSat observations.

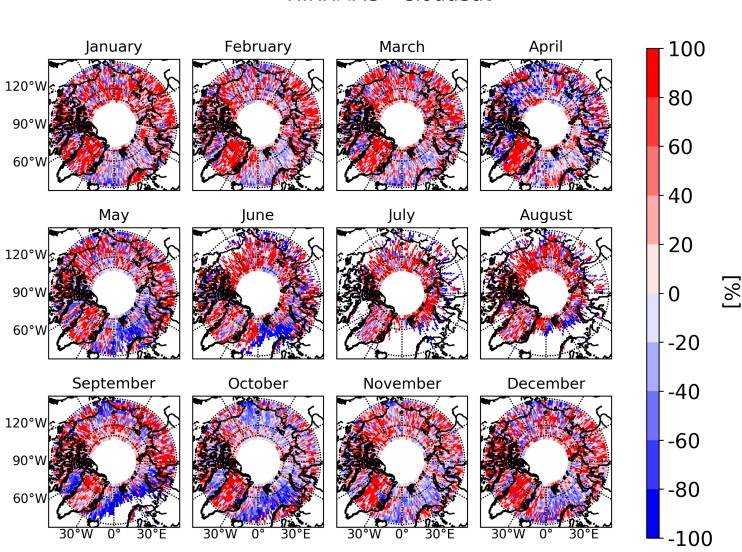

**Figure A4.** Difference of the monthly mean snowfall rates [%] of HIRHAM5 and CloudSat. Here with red colors HIRHAM5 is showing higher rates, whereas with blue colors CloudSat is observing higher rates.

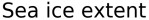

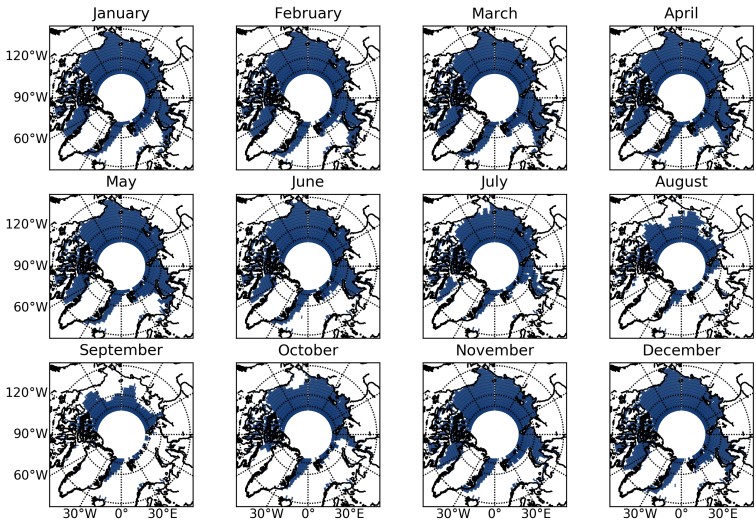

**Figure A5.** The monthly sea ice extent is defined from space-borne observations of AMSR-E/AMSR2 on a 6.25 km grid (Spreen et al., 2008) for the studied period.



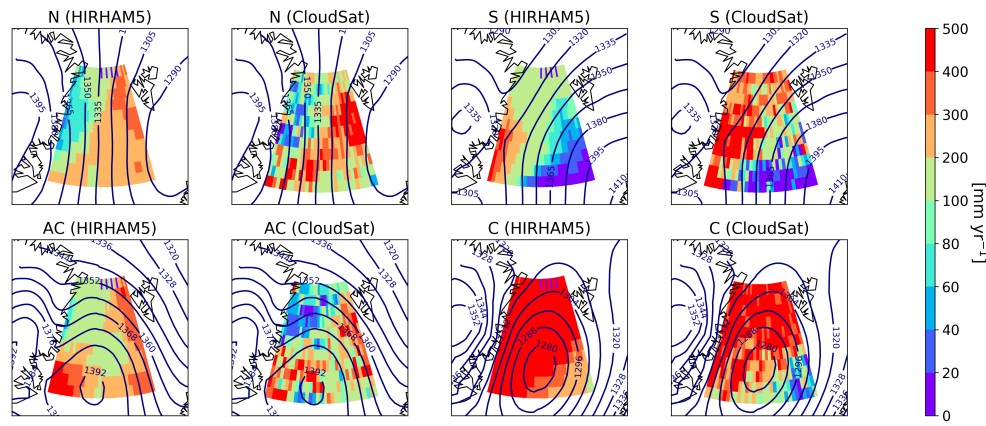

**Figure A6.** Snowfall rate composited to different CWTs, N (including N, NE, and NW) S (S, SE, and SW), C and AC, for the area of 20° W-10° E between latitude bands of 70°-81° N for both HIRHAM5 and CloudSat (the colored sub-region is according Fig. 3d). The mean 850 hPa geopotential height (gpdm) associated with the CWTs is also shown as blue contour lines

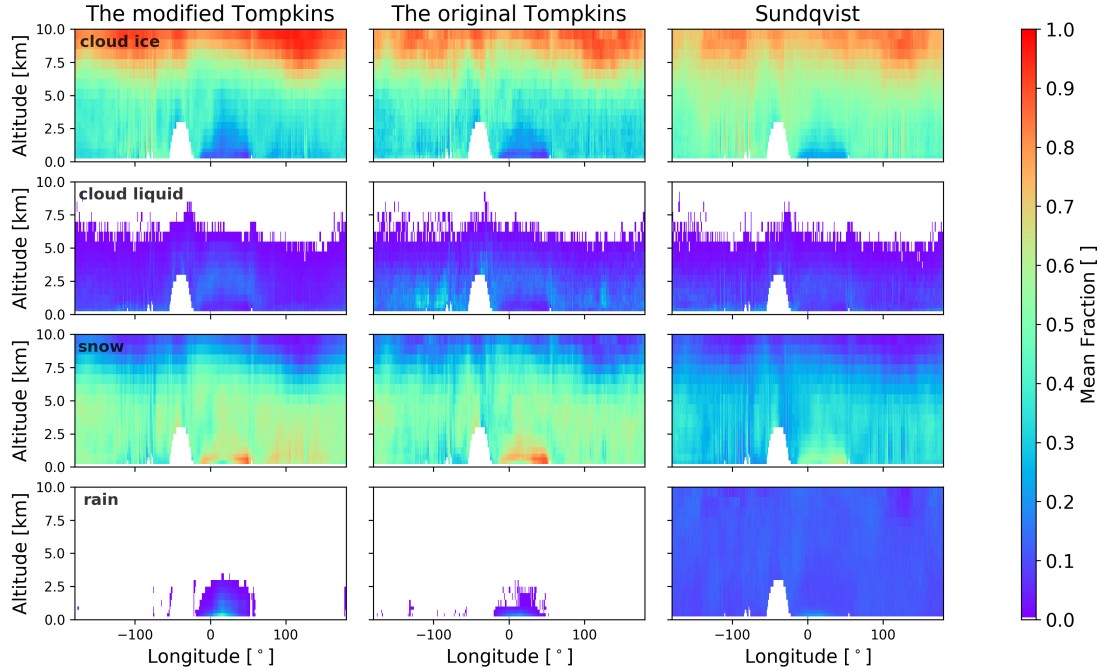

**Figure A7.** A mean fraction of reflectivity contributed by the different hydrometeors as a function of altitude for a latitude band of $72°$-$73°$ N in winter(DJF) for the different microphysical schemes, the modified Tompkins (left panel), the original Tompkins (middle panel), and Sundqvist (right panel) schemes, respectively.

*Author contributions.* S.C, A.R., and A.vL conceived and designed the analysis. A.vL collected data and performed the overall comparative analysis, D.Z. provided the CloudSat data and the derived results, M.M. performed the PAMTRA simulations, and S.C., A.R., and I.C. contributed to the interpretation of the results. A.vL took the lead in writing the manuscript with input from all authors. All authors provided critical feedback and helped shape the research, analysis, and manuscript. S.C. and A.R. supervised the research.

850   *Competing interests.* In this study there is no competing interest.





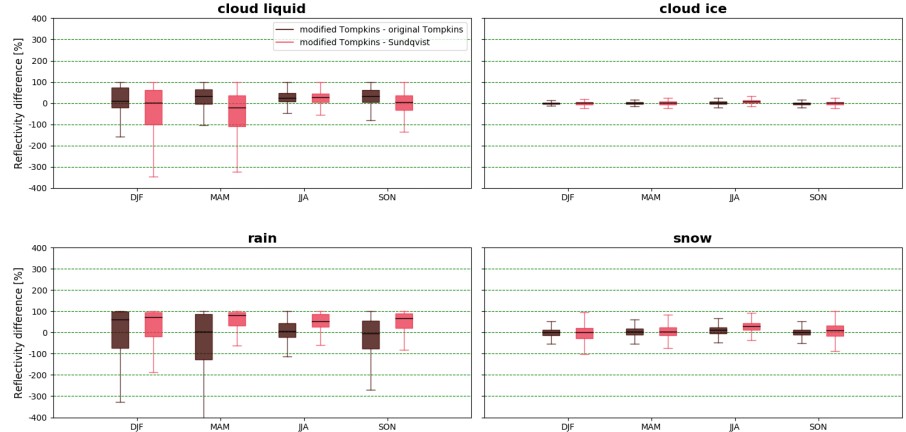

**Figure A8.** The box plot shows the differences in percents between the modified Tompkins scheme in respect to the two other schemes for different hydrometeors distributed seasonally.

*Acknowledgements.* We gratefully acknowledge the funding by the Deutsche Forschungsgemeinschaft (DFG, German Research Foundation) – Projektnummer 268020496 – TRR 172, within the Transregional Collaborative Research Center "ArctiC Amplification: Climate Relevant Atmospheric and SurfaCe Processes, and Feedback Mechanisms (AC)3". AvL is additionally funded by the Academy of Finland postdoctoral
855    scholarship (333901). We are thankful for the help of Ines Hebestadt with the HIRHAM5 runs, Dr.Benjamin Segger with ERA-Interim analysis, Dr.Tobias Marke with providing the CWT data for the two regions, Dr. Philip Rostosky deriving the monthly sea ice extent and both Dr. Christine Nam and Dr. Vera Schemann for the consultation in the HIRHAM5 model runs.