# Peer review of "Evaluating seasonal and regional distribution of snowfall in regional climate model simulations in the Arctic"

_Atmospheric Chemistry and Physics, 2021_

## Author Comment (AC1)

Reply to Referee#1 on Manuscript # acp-2021-1064 in ACPD: "Evaluating seasonal and regional distribution of snowfall in regional climate model simulations in the Arctic" by Lerber et al.

We would like to express our sincere appreciation to the reviewer for the valuable comments and corrections on our manuscript and the opportunity to revise and improve the manuscript.

We have included the reviewer comments below in bold and italic, and responded to them individually following the numbering. The individual comments and responses are followed by the revised text, with changes highlighted with colors: deletions in red and additions in blue.

***1. The major comment for this manuscript is that the authors could consider some objective metrics in the evaluation, e.g., spatial correlation, Taylor skill score.***

Thank you for your comment, but we have to take the poor sampling of CloudSat into account, which limits a throughout quantitative evaluation. Therefore, we believe that our presented approach to quantitatively evaluate HIRHAM with CloudSat is solid and the best one could do. Saying this, we present the spatial distribution (Figs. 3, 6, 8), means (Figs. 4b, 5), and frequency distribution (Fig.4a).

Here is our rationale why we came to concise to the present version. As stated, the manuscript studies the differences with two approaches, i.e. the assessment of surface snowfall rate (observation-to-model) and the radar reflectivity factor profiles (model-to-observation).

In the part of surface snowfall rates, the results are mostly compared to the study of Edel et al. (2020), where the analysis was performed by looking at the distribution of snowfall rates, distributed yearly or seasonally, and studying differences of mean snowfall rates as we presented them in this current version of the manuscript. We want to report our findings comparable to theirs to be able to build the link between these two studies. Additionally, with surface snowfall rates we compare the model and observations, how well model qualitatively reproduces the CWTs.

Due to CloudSat's long revisiting time, daily skill scores are not providing meaningful comparison as it was shown in Souverijns et al. (2018). For the monthly and yearly means, the spatial differences and frequency distribution are a well-suited evaluation. According to your comment, we include now additionally the RMSE:

In lines 341-343: "Though model and observations show similar spatial distributions, distinct spatial differences occur (Fig. 3c), and e.g. root-mean square error in the yearly surface snowfall rates is high with 148 mmyr$^{-1}$ between HIRHAM5 and CloudSat, and 175 mmyr$^{-1}$ between ERA-Interim and CloudSat."

In the part, where we investigate the differences in radar reflectivity factor profiles and CFTDs, the obviously largest differences occurred due to the small reflectivity portion of the too small ice particles and how, mainly due to this, in general, model and observations have highest reflectivity quantities in different altitudes, although clearly by regionally and spatially in the Arctic, the model reproduced the snowfall well as seen e.g. in Figure 8. Spatial correlation scores gave unjustified poor scores due to this above-mentioned difference. Therefore, we stay with these shown differences.

***2. Lines 105-106: Why was 70N selected as the boundary of two rings? Please justify it***

Many literature takes the 70°N latitude as a measure of the central Arctic or "mean Arctic". Examples are: Wang et al. (2012, doi:10.1155/2012/505613), Screen et al. (2011, doi:10.1029/2011JD015847), Koenigk et al. (2014,10.1007/s00382-013-1821-x), Bintanja (2018, doi: 10.1038/s41598-018-34450-3).

The additional description has been added:
"The two rings are separated to clarify the different characteristics of the southern and northern regions, where the 70°N defines the central Arctic boundary and also coarsely separates the Arctic Sea regions from the Arctic continental regions."

**3. Lines 128-130: I might not agree with this statement. Simulation uncertainty comes from many aspects, and microphysics parameterization is only one of them. Boundary layer parameterization can also significantly influence the model dynamics and then influence the snowfall simulation. If the authors did not conduct the sensitivity test on model physics schemes, it is not suitable to give this statement.**

We suggest a modification to the text:

"Therefore, it is assumed that the differences between the modeled snowfall and observations are in lesser degree related to the simulated large-scale flow but mostly caused by the ECHAM5 boundary layer and microphysical parameterization employed in HIRHAM5 and observational uncertainties."

**4. Section 2.3: Please provide the quantitative uncertainties of the two CloudSat products.**

The 2B-GEOPROF-product output is reflectivity factor profile. The sources of measurement uncertainty include uncertainty in the absolute radiometric calibration and measurement noise. The noise characteristics of the CPR vary with signal strength. It is estimated in Wood et al. that the resulting uncertainties range from 3 dBZ for a reflectivity of -30 dBZ to about 0.1 dBZ for reflectivities above -10 dBZ. Calibration errors, which would result in a bias in the measured reflectivities, are expected to be less than 2 dB based on a prelaunch calibration error budget (Tanelli et al., 2008), but the value of this bias is basically unknown and typically not considered. We added in line 168:

"The minimum detectable reflectivity is dependent on, e.g. cloud cover, seasonal changes in temperature, surface type, and atmospheric attenuation, typically varying by ~1 dB over the globe in the range from -30.9 to -29.9 dBZ (Tanelli et al., 2008) and the measurement uncertainties related to noise range from 3 dBZ for a reflectivity of -30 dBZ to about 0.1 dBZ for reflectivities above -10 dBZ (Wood and L'Ecuyer, 2018).

The uncertainty of snow profile product, 2C-SNOW-PROFILE, was expressed in Edel et al. 2020 as a relative uncertainty, which is the ratio between the mean single surface snowfall rate uncertainty and the surface snowfall rate, ranging from 1.5 to 2.5, with higher values associated to complex topography and high frequency of mixed phase precipitation. There are a few studies, which have estimated the accuracy of the product respect to weather radar estimated snowfall rate (Cao et al. 2014, Norin et al. 2105) with very similar results. The product has a good detectability of light snow (snow water equivalent less than 1 mm h$^{-1}$), however limited ability to retrieve at the higher end of snowfall intensity distribution (> 1 mm h$^{-1}$). We added in line 193

"Thus, we are confident to use the output of 2C-SNOW-PROFILE product as the ground truth, though acknowledging the relevant unreliability stemming from the uncertainties in observed reflectivities, the used retrieval parameters and it's a priori assumptions (Edel et al., 2020). The product has shown a good detectability of light snow (snow water equivalent less than 1 mm h$^{-1}$), however limited ability to retrieve at the higher end of snowfall intensity distribution (> 1 mm h$^{-1}$) when compared to weather radar estimated surface snowfall rate (Cao et al. 2014, Norin et al. 2015). The relative uncertainty of the product increases with complex topography and higher frequency of mixed phase precipitation (Edel et al. 2020)."

**5. Line 144: "Sect. A" -> "Appendix A"**

Corrected.

**6. Line 280: "East regions" -> "East region"**

Corrected.

**7. Line 294: "mmyr−1" -> "mm yr−1", please correct others throughout the manuscript.**

Corrected.

**8. Section 4.1: It is better to show the locations of Greenland, Barents, Kara Seas, etc.**

The names of the Seas are now written in Figure 1 to ease the reading of the Section 4.1.

**9. Lines 400-401: It would be better to show the results in other seasons in appendix or supplement.**

The CFTDs images for other seasons are added to Appendix with other additional images. And text is added in the line 401

"For investigating the differences in the vertical reflectivity structure between the different regions we focus on the winter season (DJF) which cover snowfall rates of approximately 30% over all seasons. Furthermore, we reduce problems related to mixed-phase conditions as temperatures are generally low. The other seasons are shown in the Appendix B."

**10. Section 5.1: Please discuss the differences more quantitatively.**

We have added quantified occurrences and other values to the section in lines 417- 426.

Due to the lower occurrence (< 0.1%) of cold temperature reflectivities, reflectivities at warmer temperatures are relatively more frequent in HIRHAM5 than in CloudSat observation, with occurrences of > 0.8 % for HIRHAM5 and with occurrences between 0.6 - 0.8 % for CloudSat. However, HIRHAM5 is able to reproduce regional differences seen by Cloudsat correctly. Enhanced reflectivity related to the snow mode (- 10 and 5 dBZ) occurs at the warmest temperature in the North Atlantic (around -10°C) in both observations and model, similar at slightly warmer temperature in the Kara Sea regions. In the Chukchi Sea, occurrences (0.4 - 0.8 %) are confined to a narrow temperature range between -20 and -35°C, while in the Laptev Sea the distribution broadens to colder temperature again in both observations and simulations. In the Chukchi Sea, HIRHAM5 can also reproduce the increased reflectivity occurrence (0.6%) around -20°C in the lower latitude region compared to the higher latitude region. The strongest difference between the observed and simulated CFTDs is visible for Greenland where the simulations show reflectivities at much warmer temperatures (-20 to -10°C) and higher reflectivities (0-10 dBZ) consistent with the overestimation in snowfall rate by HIRHAM5 discussed before.

**11. Figures 4 and 9: Please conduct significance test of difference for (b).**

The student t-test is performed to the differences with random samples (10% of the total amount) of the defined difference distributions. The results show that the shown median difference is statistically robust. The tables for the results are shown below for you, and the text is added to captions of Figure 4b and Figure 9b:

Figure 4b:
"The significance of the median difference for both HIRHAM5 and ERA-Interim compared to CloudSat observations is shown to be statistically robust for all seasons performing the student t-test with random samples (10% of the total amount) of the observed difference distributions."

Figure 9b:
"The significance of the median difference for both original Tompkins and Sundqvist schemes to modified Tompkins scheme is shown to be statistically robust for all seasons performing the student t-test with random samples (10\% of the total amount) of the modeled difference distributions.

In the tables, the t-value quantifies the difference between the population means. Here, the other population is the 10% random samples of the difference distribution, and the other is the total distribution. The p-value is the probability of obtaining a t-value.

For boxplot 4b:

| | HIRHAM5 | | ERA-I | |
|---|---|---|---|---|
| | t | p | t | p |
| **DJF** | 0.621 | 0.535 | -0.507 | 0.612 |
| **MAM** | 0.332 | 0.740 | -0.563 | 0.574 |
| **JJA** | 0.205 | 0.838 | -0.092 | 0.926 |
| **SON** | 0.336 | 0.737 | -0.507 | 0.612 |

and for the boxplot 9b

| | Original Tompkins | | Sundqvist | |
|---|---|---|---|---|
| | t | p | t | p |
| **DJF** | 0.401 | 0.689 | -0.444 | 0.657 |
| **MAM** | 0.338 | 0.735 | -0.577 | 0.564 |
| **JJA** | 0.032 | 0.974 | 0.388 | 0.698 |
| **SON** | -0.452 | 0.651 | 0.392 | 0.695 |

**12. Figure 7: "CFTD" -> "CFTDs", What's the meaning of "86500 - 6.6·10$^6$" in the caption?**

The sentence is modified to make it clearer:

"the normalization is done by the sum of total hits, which varies from region and season, but typically the number of hits is ranging between 86500 - 6.6·10$^6$."

**13. Lines 456-457: Please show the locations of the North Atlantic region, the East Siberian Sea and the Beaufort Sea in Figure 9.**

The regions with approximate longitude degrees are now written to the text in in lines 456-457. The modified text is:
"The highest reflectivity values due to rain particles are concentrated in the North Atlantic region (20°W - 10°E) and some higher values are also modeled in the East Siberian Sea (150°E - 180°E)  and the Beaufort Sea (150°W - 130°W)."

Reference:

Norin, L., Devasthale, A., L'Ecuyer, T. S., Wood, N. B., and Smalley, M.: Intercomparison of snowfall estimates derived from the CloudSat Cloud Profiling Radar and the ground-based weather radar network over Sweden, Atmos. Meas. Tech., 8, 5009–5021, https://doi.org/10.5194/amt-8-5009-2015, 2015.

Qing Cao, Yang Hong, Sheng Chen, Jonathan J. Gourley, Jian Zhang, and Pierre E. Kirstetter, "Snowfall Detectability of NASA's CloudSat : the First Cross-Investigation of ITS 2c-Snow-Profile Product and National Multi-Sensor Mosaic QPE (NMQ) Snowfall Data," Progress In Electromagnetics Research, Vol. 148, 55-61, 2014,doi:10.2528/PIER14030405

---

## Author Comment (AC2)

Reply to Referee#2 on Manuscript # acp-2021-1064 in ACPD: "Evaluating seasonal and regional distribution of snowfall in regional climate model simulations in the Arctic" by Lerber et al.

We are extremely grateful and naturally very happy for the reviewer#2 to make such an encouraging review. We appreciate for the comments to improve the overall readability of the paper and to indicate us the parts that needed further explanation.

We have included the reviewer comments below in bold and italic, and responded to them individually following the numbering. The individual comments and responses are followed by the revised text, with changes highlighted with colours: deletions in red and additions in blue.

**1. Line 239-240: "Multiple scattering …. not considered in these computations", does this mean that the model-to-observation process does not include multiple scattering? If so, how does this affect the comparison?**

Yes, correct. The scattering computations in PAMTRA are not considering the multiple scattering effect, but the single scattering approximation is assumed to be valid. As stated in the lines 239-240, the effect of multiple scattering "…can be approximately 1 dB in snowfall with reflectivity values greater 10-15 dBZ". These numbers are from the study of Matrosov and Battaglia, 2009. There it is also stated that the reflectivity enhancement due to multiple scattering can be as high as 5 dB in heavy stratiform snowfalls in W-Band and that multiple scattering effects counteracts with signal attenuation. Thus, in the two comparisons we have in model-to-observation space, i.e. the CFTDs (Figure 7.) and reflectivity profiles (Figure 8.), the multiple scattering would influence particularly at the warmer temperatures and lower altitudes, respectively, where assumably the snowfall intensity is heavier.

In case of the CFTDs, we study the temperature region up to -10°C. It can be assumed that most likely the multiple scattering should not play a major role as the reflectivity region is mostly below 0 dBZ, and according to the results, even considering the attenuation, the model sees higher reflectivities in these -15 - -10 °C temperatures. However, with the study of mean reflectivity profiles at low altitudes the multiple scattering may be the explanation, why it seems that observed reflectivities at the surface are higher than the model estimates when the attenuation is considered. And reasoning for this is not that the model overestimates the attenuation as assumed in the manuscript, but that multiple scattering counteracts the attenuation in the observations. With this in mind, we have changed the sentences in lines 506-508:

"Generally, it seems that the modeled attenuation is higher than actually seen in the observations, especially during summer months. This difference could also (at least partly) be explained with multiple scattering effects which would counteract attenuation (Matrosov and Battaglia, 2009). However, without considering attenuation in the simulations, the model overestimation increased by more than 10 - 20%."

**Commented [SC1]:** I am still unhappy with this sentence – how can you know? Or maybe I forgot an explanation

**Commented [vLA(2R1]:** In the mean modeled profiles with the attenuation, the observed CloudSat reflectivites were increasingly higher than modeled ones when closer to the surface. If attenuation was not considered HIRHAM5 modeled higher reflectivites closer to the surface.

**2. Section 3.2: suggest to add more details on the procedure of using the Jenkinson-Collison method for circulation weather type classification.**

The revised chapter is now:

In order To better identify reasons for potential deviations between observations and HIRHAM5, we also composite snowfall maps for different distinguishable weather regimes and evaluate the model output to observations in each regime separately as performed in Akkermans et al. 2012. Separating the daily modeled and observed snowfall rates according to an external parameter allows us to identify possible systematic model biases related to synoptic processes. In this study, we chose to investigate regimes of large-scale atmospheric circulation classified by strength, direction, and vorticity of the geostrophic wind.

We selected two sub-regions of the northern North Atlantic around Svalbard (Fig. 3d) covering the latitude band of 70° N - 81° N. The regions East (40° E - 10° E) and West (20° W - 10° E) are considered such that the East region is directly north of Scandinavia and includes Svalbard while the West region avoids land regions and is placed between Greenland and Svalbard. Both areas are characterized by high synoptic variability with frequent cyclone passages. The regime classification was performed with ERA-Interim 6-hourly 850 hPa geopotential height and shear vorticity for the studied period with the methodology of Jenkinson-Collison (Jenkinson and Collison, 1977, Philipp et al. 2016). The geopotential at 850 hPa is used to avoid topographic and boundary layer effects. The Jenkinson – Collision method is an automatic classification scheme (Philipp et al. 2016), where the geostrophic wind speed and vorticity at high/low central pressure are assessed in horizontally and isotropically arranged grid points and based on threshold values are set to eight exclusionary directional classes according to compass points (N, NE, E, SE, S, SW, W, and NW) and two vorticity circulation regimes (cyclonic (C) and anticyclonic (AC) (Akkermans et al. 2012)).

The daily regime classification of the two sub-regions (West/East) is specified with the software package *cost733class* of the COST Action framework (Philipp et al. 2016) and the occurrence of each regime for both sub-regions are determined. Clearly, the northerly, southerly, and both vorticity classes are by far most frequent with an occurrence of 16% (25%), 12% (10%), 31% (38%), and 34% (32%), respectively for the East (West) sub-region. To simplify the analysis, the less frequent NE (6 % (7%)) and NW (9% (6 %)) were added to the northern regime typically representing situations when cold Arctic air masses move southward. Similarly, SW (7% (10%)) and SE (8% (3%)) were added to the southern cluster which is a typical situation for warm air intrusions into the Arctic. Finally, we divided the modeled and observed daily mean snowfall rates to these four regimes and calculated the contribution to the yearly mean snowfall rate to each regime separately.

***3. Caption of Figure 7: The number at the end didn't show up correctly in the pdf text.***

Corrected.

***4. Figure 9: Panel c was not labeled.***

Corrected.